# GASDU: Gauss–Southwell Dynamic Update for Efficient LLM Fine-Tuning

## Abstract

Parameter-efficient fine-tuning (PEFT) is crucial for adapting large language models (LLMs), yet existing methods trade off accuracy, latency, and compute: some add inference-time modules, others fix a static parameter set that can drift from evolving gradients, and dynamic variants can be costly. We propose **GA**uss–**S**outhwell **D**ynamic **U**pdate (GASDU), which performs *periodic Gauss–Southwell-$k$* selection: every $M$ steps it uses the current gradients to select the $k$ largest-magnitude coordinates and updates only those entries while reusing the mask until the next refresh. The Top-$k$ selection is implemented in a streaming, tile-wise way to avoid materializing dense gradients, making the amortized refresh cost negligible. Theoretically, under a local Polyak–Łojasiewicz condition, we prove that GASDU enjoys a linear convergence rate scaled by a measurable gradient-retention factor and show that the factor degrades sublinearly within each refresh window. This sublinear decay implies that a moderate $M$ can maintain a high retention factor, which in turn explains GASDU's near–full–fine-tuning behavior. Empirically, GASDU sustains high retention between refreshes at an extreme parameter budget (0.01%) and consistently outperforms strong PEFT baselines and closely tracks or exceeds full fine-tuning across diverse commonsense and arithmetic reasoning benchmarks and LLMs (LLaMA-2/3 and GPT-OSS-20B).

## 1 Introduction

Adapting large pretrained language models (LLMs) to specialized tasks, such as biomedical text mining, financial analysis, and legal document review, has achieved remarkable success (Ruan et al., 2025; Fang et al., 2025; Lu et al., 2025). This progress relies on fine-tuning, where models are adjusted with domain-specific data to bridge the gap between general pretraining and downstream requirements (Hu et al., 2022; Sung et al., 2021; Dettmers et al., 2024; Guo et al., 2021; Han et al., 2024; Liao et al., 2023). However, as model scale expands into the billions of parameters, conventional full-parameter fine-tuning has become prohibitively expensive. Parameter-Efficient Fine-Tuning (PEFT) methods have emerged as a solution, enabling adaptation by updating only a small fraction of the model parameters (Han et al., 2024).

While PEFT makes adaptation practical at scale, many designs entail trade-offs. Some methods constrain updates with low-rank adapters, limiting optimization flexibility; some fix a static sparse subset of weights, which can drift as gradient directions evolve; and some dynamically reallocate parameters to better follow the optimization geometry, but often rely on dense-gradient materialization or extra passes that erode efficiency (Han et al., 2024). These limitations motivate a latency-neutral, dynamically refreshed update rule that tracks evolving gradient directions without significantly inflating computational and memory cost.

We introduce **GA**uss–**S**outhwell **D**ynamic **U**pdate (GASDU), a novel PEFT strategy that efficiently approximates the ideal dynamic update. GASDU operates by performing *periodic Gauss–Southwell-$k$* selection (Nutini et al., 2015) over the model's parameters: every $M$ steps, it leverages the current gradients to identify the $k$ parameters with the largest gradient magnitudes and updates only those entries until the next refresh. To keep selection cost low, we compute Top-$k$ coordinates via a *streaming, tile-wise* reduction with a small $O(k)$-sized candidate pool, discarding tiles immediately and never materializing the full per-weight gradient matrix in high-bandwidth memory. By

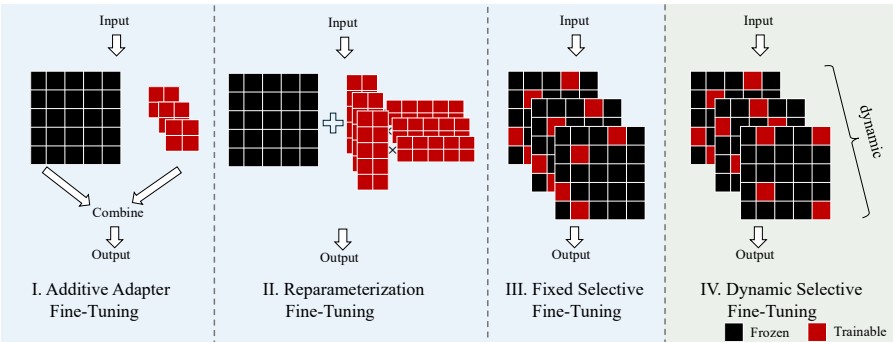

Figure 1: Illustration of additive adapter methods, reparameterization approaches, fixed selective fine-tuning methods, and dynamic selective method (to which GASDU belongs). Whereas the first three either attach extra modules, tune reparameterized components, or update a static subset of parameters, GASDU applies a dynamic sparse linear update directly to the pre-trained parameters.

refreshing periodically, GASDU amortizes the selection cost and dynamically tracks the evolving optimization direction without extra passes that are common to prior dynamic selection methods. By design, GASDU preserves the original architecture, adds no inference latency, and sidesteps the high costs of prior dynamic methods while maintaining training stability.

From a theoretical perspective, we prove that GASDU achieves a linear convergence rate close to that of full-parameter fine-tuning (Section 4.1), under the local Polyak–Łojasiewicz (PL) condition, which is empirically verified in Section 4.3. The difference in rate arises only from the fraction of gradient norm retained by the sparse mask. Our Mask-Reuse Retention Analysis (Section 4.2) further shows that when the mask is reused for multiple steps, the retention factor decays sublinearly within each refresh period, ensuring convergence remains close to full fine-tuning. We empirically confirm that the dynamic mask consistently captures the dominant gradient norm (Section 5.2), allowing GASDU to closely follow dense optimization while updating only a tiny subset of parameters. Furthermore, the aforementioned retention-based analysis is readily applicable to other selective fine-tuning schemes.

We evaluate GASDU on diverse commonsense and arithmetic reasoning tasks under a 0.01% update budget. Across LLaMA-2-7B/3-8B and GPT-OSS-20B models, GASDU consistently outperforms leading PEFT baselines at the same sparsity and, on several tasks, surpasses full fine-tuning (Section 5.1). In terms of training efficiency, GASDU delivers a **10.64×** throughput improvement and reduces peak GPU memory to **30%** of full fine-tuning (Section 5.3). Per-iteration profiling shows the mask-refresh cost is relatively small and largely insensitive to the update budget since our streaming Top-$k$ implementation avoids materializing dense gradients. With a modest refresh period $M$, only one in $M$ steps incurs this cost, so the amortized overhead becomes negligible (Section 5.3). Overall, GASDU preserves the benefits of dynamic mask selection to match or exceed the predictive performance of full fine-tuning while achieving substantial speedup and memory savings.

## 2 RELATED WORK

Existing PEFT approaches follow four main paradigms, as demonstrated in Figure 1. *Additive methods* (e.g., adapters) introduce new trainable modules while freezing the rest weights, which reduces the number of updated parameters but can increase memory usage and inference latency (Houlsby et al., 2019; Li & Liang, 2021). *Reparameterization methods*, such as LoRA (Hu et al., 2022), constrain updates to a fixed low-rank subspace, preserving inference speed but restricting optimization flexibility (Zhang et al., 2023; Dettmers et al., 2024). *Fixed Selective methods* (e.g., SIFT (Song et al., 2023)) update only a predetermined subset of parameters throughout training (Ben Zaken et al., 2022; Guo et al., 2021; Sung et al., 2021). The reliance on a fixed parameter subspace poses a common limitation, namely the risk of converging to suboptimal solutions, as the set of optimal parameters to update can shift during optimization. This limitation has motivated the development of *Dynamic Selective methods*, which relaxes a fixed mask by changing which base weights are updated during training. RigL alternates prune/regrow based on magnitude-gradient signals, inserting

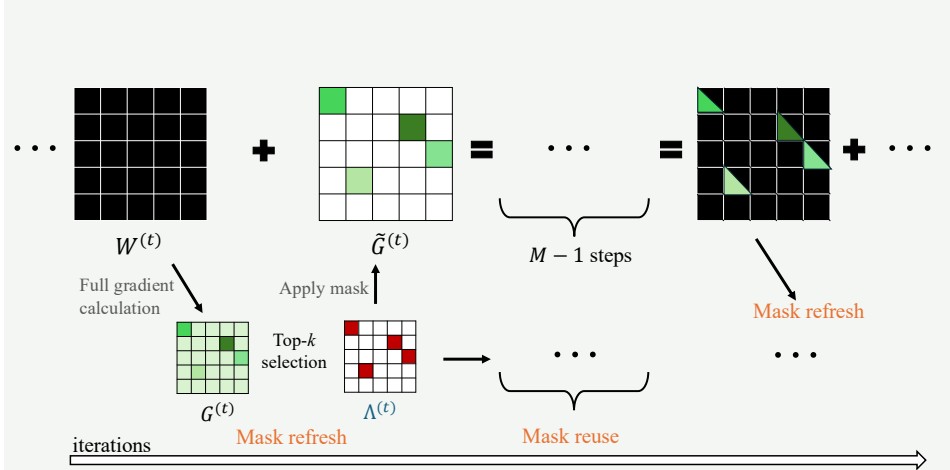

Figure 2: Workflow of the GASDU fine-tuning algorithm. At each training iteration, the sparse mask is either reused from the previous step or refreshed every $M$ steps via Top-$k$ selection of the full gradient. When the mask is reused, the model applies a sparse parameter update restricted to the current mask without performing dense back-propagation.

periodic extra updates to change the support (Evci et al., 2020). For LLMs, SpIEL scales sparse fine-tuning by iterating update-prune-regrow cycles, where regrowth uses accumulated gradients or SM3 momenta; it improves over LoRA at comparable runtime but incurs non-negligible evaluation/selection overheads (Ansell et al., 2024). Dynamic Subset Tuning (DST) similarly optimizes a small, moving subset of existing parameters rather than a fixed mask (Stahlberg et al., 2024). In addition, sampling/structured/hybrid designs (e.g., LISA, S$^2$FT, SLTrain) add implementation and tuning complexity (Pan et al., 2024; Yang et al., 2024; Han et al., 2024). Among these, S$^2$FT enforces structured sparsity patterns to improve hardware efficiency and cross-task generalization in large-scale LLM fine-tuning. SparseLoRA further combines low-rank adapters with contextual sparsity, activating LoRA modules only on a subset of tokens or positions to reduce both training and inference cost while remaining in the additive-PEFT paradigm (Khaki et al., 2025). Orthogonal to parameter selection, GaLore compresses gradients via low-rank projection before the optimizer update, substantially reducing optimizer and gradient memory while still performing full-parameter updates rather than selective fine-tuning (Zhao et al., 2024).

Despite extensive empirical progress, to the best of our knowledge, general convergence guarantees for nonconvex LLM fine-tuning with changing sparsity patterns remain scarce. When analyses do appear, they are restricted to simplified or surrogate settings and do not quantify how design parameters such as the refresh cadence $M$ and rotation/sampling schedules affect objective decrease. In particular, prior work does not provide rate statements or monotone-decrease guarantees tied to a measurable control quantity that explains when and why dynamic selection maintains near–full-fine-tuning behavior. This gap leaves practitioners without principled guidance on how refresh frequency and selection granularity should scale with budget and model regime.

**Positioning of GASDU.** To close this gap, GASDU employs gradient-driven *periodic* updates that avoid auxiliary modules and extra passes, preserving inference latency while keeping refresh overhead amortized and small. More importantly, it provides, to our knowledge, the first retention-based convergence analysis for selective PEFT under a *verifiable* local Polyak–Łojasiewicz (PL) condition. The theory makes explicit how $M$ (refresh frequency) and $k$ (update granularity) influence the convergence rate via the measurable gradient-retention factor $\alpha$, yielding principled guidance for setting these hyperparameters across budgets and model regimes. This combination of practical efficiency and explicit design rules distinguishes GASDU as both effective and theoretically grounded.

## 3 METHOD

To achieve efficient fine-tuning while preserving convergence speed and task performance, we propose Gauss-Southwell Dynamic Update (GASDU), a method that dynamically sparsifies parameter updates. The workflow is illustrated in Figure 2.

We fine-tune a pre-trained parameter matrix $W \in \mathbb{R}^{m \times n}$ by updating only a small, dynamically chosen subset of its entries. Let $\Lambda^{(t)} \in \{0, 1\}^{m \times n}$ be a binary mask that selects exactly $k \ll mn$ coordinates at iteration $t$. The *masked* update on the selected entries is

$$w_{ij}^{(t+1)} = w_{ij}^{(t)} - \gamma \nabla_{w_{ij}} f(W^{(t)}), \qquad \text{for all } (i, j) \text{ with } \Lambda_{ij}^{(t)} = 1, \tag{1}$$

while all other coordinates remain frozen. Equivalently, in the unified masked-gradient form used in our analysis (cf. Section 4),

$$W^{(t+1)} = W^{(t)} - \gamma \left( \Lambda^{(t)} \odot \nabla f(W^{(t)}) \right),$$

which coincides with full-parameter gradient descent when $\Lambda^{(t)} \equiv \mathbf{1}$ (see Eq. (3)).

Every $M$ steps we *refresh* the active set by taking the Top-$k$ coordinates of the current full gradient magnitude:

$$\Lambda^{(t)} = \text{TopK}\left( \nabla f(W^{(t)}), k \right), \quad \text{where} \quad \text{TopK}(G, k)_{ij} = \begin{cases} 1, & |g_{ij}| \text{ is among the } k \text{ largest in } |G|, \\ 0, & \text{otherwise.} \end{cases} \tag{2}$$

Between refreshes, the mask is reused, so the update (1) operates on a fixed set of coordinates for $M$ iterations. This "periodic Gauss–Southwell-$k$" rule lets the method follow the dominant descent directions without introducing any inference-time modules.

---

**Algorithm 1** GASDU: Gauss–Southwell-$k$ Dynamic Update

---

**Require:** Pre-trained $W^{(0)}$, loss $f$, step size $\gamma$, sparsity $k$, refresh period $M$, total iterations $T$
1: **(init)** Compute $G^{(0)} \leftarrow \nabla f(W^{(0)})$; set $\Lambda^{(0)} \leftarrow \text{TopK}(G^{(0)}, k)$
2: **for** $t = 0$ **to** $T - 1$ **do**
3:     **if** $(t+1) \bmod M = 0$ **then**               ▷ streaming and tile-wise mask refresh
4:         $G^{(t)} \leftarrow \nabla f(W^{(t)})$
5:         $\Lambda^{(t)} \leftarrow \text{TopK}(G^{(t)}, k)$               ▷ install the fresh mask immediately
6:     **else**
7:         $\Lambda^{(t)} \leftarrow \Lambda^{(t-1)}$                          ▷ reuse mask
8:     **end if**
9:     $\tilde{G}^{(t)} \leftarrow \Lambda^{(t)} \odot \nabla f(W^{(t)})$               ▷ masked gradient (computed sparsely)
10:    $W^{(t+1)} \leftarrow W^{(t)} - \gamma \tilde{G}^{(t)}$
11: **end for**
12: **return** $W^{(T)}$

---

**Speed and memory optimizations.** We reduce computational time and memory of mask refresh by *streaming* the per-weight gradient in small `bf16` tiles instead of materializing the full $m \times n$ matrix. For each tile, we compute the magnitudes, select the tile's Top-$k$ entries, merge into an $O(k)$-sized candidate pool with `fp32` accumulations, and immediately discard the tile. As a result, the full gradient is never written to or read from high-bandwidth memory (HBM), eliminating $\Theta(mn)$ traffic and keeping peak working memory proportional to $k$ rather than $mn$. The refresh is integrated into the backward pass of the current minibatch, reusing existing gradients and activations for streamed Top-$k$ selection, thus eliminating the need for an extra forward/backward pass. Because a sweep occurs only once every $M$ steps, the amortized cost over $T$ iterations is $\mathcal{O}\left(\frac{T}{M}mn + Tk\right)$ instead of $\mathcal{O}(Tmn)$. Between refreshes, computation is restricted to the $k$ active coordinates, and updates use blockwise in-place accumulation over the active index set, avoiding large temporaries and further reducing memory traffic.

**Parallel training compatibility.** Because all operations in GASDU (streaming Top-(k), sparse updates, and commit) act on local linear projections and their gradients, the method naturally extends to tensor-parallel (TP) setups by selecting and updating Top-(k) entries independently on each

shard. In Fully Sharded Data Parallel (FSDP) training, the sparse update vectors and mask indices are treated as ordinary trainable parameters that can be sharded or replicated alongside the frozen backbone, so integration with TP/FSDP requires only standard configuration rather than any algorithmic changes.

# 4 THEORETICAL ANALYSIS

In this section, we formalize GASDU as masked gradient descent and show that, under a local Polyak–Łojasiewicz (PL) condition, it enjoys linear convergence with a rate scaled by a measurable gradient-retention factor (the fraction of $\ell_2$ gradient energy captured by the active mask). We further derive a lower bound of the retention factor which shows it decays sublinearly in each refresh period.

## 4.1 CONVERGENCE ANALYSIS

We first show that full-parameter fine-tuning is a special case of GASDU formally. For the parameter matrix $W \in \mathbb{R}^{m \times n}$, a unified update rule that encompasses both full-parameter gradient descent (GD) and GASDU (Eq. (1)) at iteration $t$ is:

$$W^{(t+1)} \;=\; W^{(t)} - \gamma \Lambda^{(t)} \odot \big(\nabla f(W^{(t)})\big), \tag{3}$$

where $\Lambda^{(t)} \in \{0,1\}^{m \times n}$ is an iteration-dependent binary mask matrix and $\odot$ denotes the Hadamard (element-wise) product. Note that in applications, we do not need to calculate the gradients of the parameters corresponding to the zero entries in $\Lambda^{(t)}$, which leads to significant computational savings by avoiding unnecessary gradient evaluations. Full-parameter GD is recovered by taking $\Lambda^{(t)} = \mathbf{1}_{m \times n}$ for all $t$, whereas GASDU uses a sparse mask obtained as $\Lambda^{(t)} = \mathrm{TopK}\big(\nabla f(W^{(t)}), k\big)$ (Eq. (2)). The gradient norm captured by $\Lambda^{(t)}$ can be measured by the gradient retention factor $\alpha_t$:

$$\alpha_t \;=\; \frac{\|\Lambda^{(t)} \odot \big(\nabla f(W^{(t)})\big)\|^2}{\|\nabla f(W^{(t)})\|^2}. \tag{4}$$

In the special case of full-parameter GD, i.e., $\Lambda^{(t)} = \mathbf{1}_{m \times n}$, the mask retains the entire gradient and $\alpha_t = 1$. For a fixed $W^{(t)}$, the quantity $\alpha_t$ depends on the budget $k$ through the top-$k$ mask: increasing $k$ enlarges the support of $\Lambda^{(t)}$, can only increase the numerator in (4), and monotonically drives $\alpha_t$ toward 1 as $k$ approaches the full parameter count. Thus, $k$ controls how much of the full-gradient norm is preserved at each step, with larger $k$ yielding masked updates that are closer to full-gradient descent and smaller $k$ trading gradient norm for memory and compute savings.

**Polyak–Łojasiewicz (PL) condition.** Let $f : \mathbb{R}^d \to \mathbb{R}$ be the empirical loss minimized during fine-tuning (e.g., cross-entropy (Goodfellow et al., 2016)). We say $f$ is $L$-smooth if $\|\nabla f(\mathbf{x}) - \nabla f(\mathbf{y})\| \leq L\|\mathbf{x} - \mathbf{y}\|$ for all $\mathbf{x}, \mathbf{y} \in \mathbb{R}^d$. Beyond smoothness, the *Polyak–Łojasiewicz* (PL) condition (Karimi et al., 2016) asserts that for some $\mu > 0$,

$$\|\nabla f(W)\|^2 \;\geq\; 2\,\mu\,\big(f(W) - f(W^*)\big), \tag{5}$$

where $W^*$ is a global minimizer. Together with $L$-smoothness, (5) guarantees linear convergence of gradient descent (Karimi et al., 2016; Liu et al., 2022). For modern LLMs, the global PL condition rarely holds because the optimization problem is high-dimensional and non-convex, so we adopt a verifiable *local* variant.

**Condition 4.1** (Local $\mu$-PL). There exists $\mu > 0$ and a neighborhood $\mathcal{S}$ of $W^*$ such that (5) holds for all $W \in \mathcal{S}$.

Empirical results that support the existence of a local PL property during full fine-tuning of LLMs is provided in Section 4.3. With Condition 4.1 and $\alpha_t$ defined in Eq. 4, we prove that our GASDU convergence linearly, with full-parameter GD as a special case.

**Theorem 4.2** (Local PL Convergence of GASDU). *Let $f : \mathbb{R}^d \to \mathbb{R}$ be $L$-smooth and $\mu$-PL on a set $\mathcal{S} \subseteq \mathbb{R}^d$. Assume $W^{(0)} \in \mathcal{S}$ and that the iterations produced by the above update rule remain in $\mathcal{S}$. Then for any stepsize $\gamma \leq 1/L$ the sequence $\{W^{(t)}\}$ satisfies*

$$f\big(W^{(t+1)}\big) - f(W^*) \;\leq\; \big(1 - \alpha_t\,\mu\,\gamma\big)\big[f(W^{(t)}) - f(W^*)\big],$$

*and consequently, if $\alpha = \inf_t \alpha_t > 0$,*

$$f\big(W^{(t)}\big) - f(W^*) \;\leq\; \big(1 - \alpha\,\mu\,\gamma\big)^t \big[f(W^{(0)}) - f(W^*)\big].$$

*Proof.* See Appendix A.1. □

**Corollary 4.1** (Gradient Descent). When $\Lambda^{(t)} = \mathbf{1}_{m \times n}$ for all $t$ (i.e. $\alpha_t \equiv 1$), Theorem 4.2 reduces to the classical PL result: $f(W^{(t)}) - f(W^*) \leq (1 - \mu\gamma)^t \left[ f(W^{(0)}) - f(W^*) \right]$.

### 4.2 MASK-REUSE RETENTION ANALYSIS

Let $t_{\mathrm{ref}} \leq t$ be the most recent refresh index and reuse the fixed mask $\Lambda = \Lambda^{(t_{\mathrm{ref}})} = \mathrm{TopK}(\nabla f(W^{(t_{\mathrm{ref}})}), k)$ for steps $s \in \{t_{\mathrm{ref}}, \ldots, t\}$. Write $g^{(s)} := \nabla f(W^{(s)})$ and $\tau_t := t - t_{\mathrm{ref}} \in \{0, \ldots, M-1\}$. Assume the iterates remain in the local PL region $\mathcal{S}$.

**Theorem 4.3** (Retention Under Mask Reuse). *If $f$ is $L$-smooth and $\mu$-PL on $\mathcal{S}$ and $\gamma \leq 1/L$, then for any $t \geq t_{\mathrm{ref}}$,*

$$\rho_t := \frac{L}{\sqrt{2}\mu} \sqrt{\frac{\gamma \tau_t}{1 - \frac{L\gamma}{2}}} \quad \Rightarrow \quad \alpha_t \geq \frac{\left[ \sqrt{\alpha_{t_{\mathrm{ref}}}} - \rho_t \right]_+^2}{(1 + \rho_t)^2}, \qquad \rho_t \leq \frac{L}{\sqrt{\mu}} \sqrt{\gamma \tau_t},$$

*where $[x]_+ := \max\{x, 0\}$.*

**Rate with reuse.** Combining Theorem 4.3 with Theorem 4.2 gives, for every stale step,

$$f(W^{(t+1)}) - f(W^*) \leq \left( 1 - \mu\gamma \frac{\left[ \sqrt{\alpha_{t_{\mathrm{ref}}}} - \rho_t \right]_+^2}{(1 + \rho_t)^2} \right) \left[ f(W^{(t)}) - f(W^*) \right].$$

*Proof.* See Appendix A.2. □

**Overall Interpretation.** Theorem 4.2 shows that the per-iteration convergence rate is $\alpha_t \mu\gamma$; thus, larger retained gradient norm (higher $\alpha_t$) yields faster linear convergence, while smaller $\alpha_t$ slows convergence in exchange for memory and compute savings. Since Eq. (4) implies that $\alpha_t$ is non-decreasing in the budget $k$, increasing $k$ strengthens the convergence rate but raises cost, whereas smaller $k$ yields cheaper but slower updates. Theorem 4.3 further quantifies how mask reuse affects this picture: the factor $\rho_t$ measures gradient drift since the last refresh and satisfies $\rho_t = \Theta(\sqrt{\gamma \tau_t})$ up to problem-dependent constants, so the lower bound on $\alpha_t$ degrades only sublinearly in the reuse length $\tau_t$. Halving either the stepsize $\gamma$ or the reuse window $\tau_t$ shrinks $\rho_t$ by a factor of $\sqrt{2}$ and tightens the bound on $\alpha_t$. In our main experiments, we fix $k$ to update roughly 0.01% of total parameters and use a moderate refresh period $M$, under which we empirically observe that $\alpha_t$ remains high with mild oscillations (Figure 4).

### 4.3 VERIFICATION OF THE LOCAL PL CONDITION

To evaluate Condition 4.1, we monitor both the loss and the gradient norm of LLaMA-3-8B on the ARC-C and NumGLUE type1 datasets throughout full fine-tuning. At each training iteration, the empirical loss $f(W)$ is computed on a held-out validation set, while $f(W^\star)$ is approximated by the minimum observed loss during training. The gradient norm $\|\nabla f(W)\|_2$ is extracted from the backpropagated gradients at the corresponding model parameters. To avoid artifacts introduced by convergence plateaus, we remove the last 50 training points from the analysis. Plotting $\|\nabla f(W)\|_2^2$ against $f(W) - f(W^\star)$ on a log–log scale directly tests the inequality above. Indeed, taking logarithms yields

$$\log \|\nabla f(W)\|_2^2 \approx \log\left( f(W) - f(W^\star) \right) + \log(2\mu),$$

implying that the points should align along a straight line with slope close to unity when the local PL condition is satisfied.

Figure 3 reveals a near-linear slope of approximately one, thereby providing strong empirical support that the optimization trajectory during fine-tuning resides in a region of the parameter space where the local PL condition holds. Similar patterns are observed in the fine-tuning procedure of other models as well.

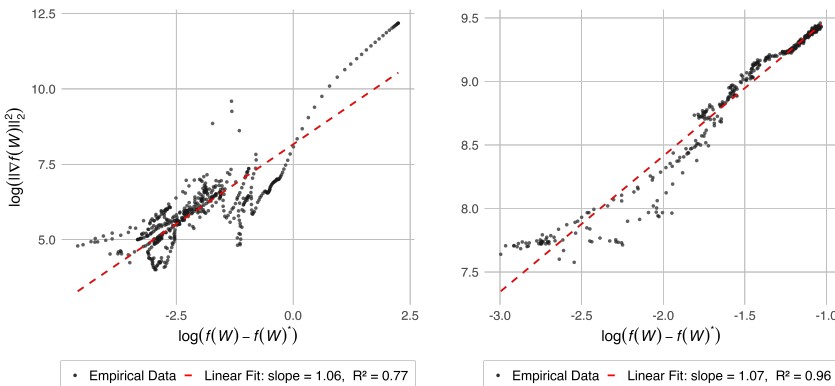

Figure 3: Empirical verification of the local PL condition for LLaMA-3-8B on ARC-C (left) and NumGLUE type1 (right). Each log–log plot demonstrates a clear linear relationship between $log(\|\nabla f(W)\|_2^2)$ and $log(f(W) - f(W^\star))$, confirming that the gradient norm remains bounded below by a positive multiple of the excess loss during full fine-tuning of LLaMA-3-8B model.

## 5 EXPERIMENTS

In this section, we evaluate the proposed GASDU method on a range of language understanding tasks. We compare GASDU with full fine-tuning and several state-of-the-art PEFT methods across diverse datasets in arithmetic reasoning and commonsense reasoning tasks. We also examine how dynamically refreshing the sparse mask during fine-tuning, rather than keeping it fixed, affects performance. To assess the robustness of GASDU, we analyze how its performance varies with different refresh period $M$. All experiments are conducted on an NVIDIA H100 GPU with 80 GB of memory.

**Models and Benchmarks.** We fine-tune and evaluate three LLMs: LLaMA-2-7B (Touvron et al., 2023), LLaMA-3-8B (Dubey et al., 2024), and GPT-OSS-20B (OpenAI, 2025). For arithmetic reasoning, we use NumGLUE (Mishra et al., 2022), which spans eight task types. We focus on six (types 1–5 and 8), excluding type6 (implicit reasoning with textual answers) and type7 (quantitative Natural Language Inference), since these require categorical or span-based outputs rather than explicit numeric predictions. For commonsense reasoning, we adopt eight established benchmarks: BoolQ, PIQA, SIQA, HellaSwag, Winogrande, ARC-Easy, ARC-Challenge, and OpenBookQA. Models are trained and evaluated separately on each dataset. More detailed descriptions are provided in Appendix B.

**Baselines.** We compare GASDU with several representative PEFT methods. LoRA (Hu et al., 2022) serves as a standard low-rank adaptation baseline. We also include recent LoRA variants: LoRA-One (Zhang et al., 2025), which uses a one-step full gradient pass to initialize a task-specific low-rank subspace, and LoRA-GA (Liu et al., 2024), which allocates non-uniform per-layer ranks using gradient-approximation statistics instead of a fixed global rank. We also include SpIEL (Ansell et al., 2024), a dynamic sparse fine-tuning method that alternates pruning and regrowth phases to adaptively maintain sparsity during training. To isolate the value of dynamic refreshing, we evaluate a static sparse variant, Fixed Mask, which selects the largest gradient coordinates once on the first batch and keeps them fixed thereafter. Full fine-tuning (full FT), which fine-tunes all parameters, serves as a reference upper bound on LLaMA-2-7B and LLaMA-3-8B. For GPT-OSS-20B, full FT was infeasible on our hardware due to memory limits, so we only report PEFT results.

**Training Pipeline.** For each PEFT method, we apply the method on the *query*, *key*, *value*, and *output* projection matrices of every transformer block and allocate the number of trainable parameters evenly across these matrices, except LoRA-GA, which uses gradient-based non-uniform rank allocation. For GASDU, the sparse mask is refreshed every $M = 50$ steps. All methods use the same budget ($0.01\%$ of total parameters). We adopt this *extreme* budget to: (i) stress-test PEFT under scarce capacity, where dynamic reallocation matters most; (ii) avoid extra capacity masking algorithmic differences in apples-to-apples comparisons; and (iii) mirror constrained-hardware deployments where memory and training cost dominate. Hyperparameters are chosen

| Model | Method | Upd.% | type1 | type2 | type3 | type4 | type5 | type8 | Avg |
|---|---|---|---|---|---|---|---|---|---|
| | LoRA | 0.01 | 27.4 | 38.4 | 50.2 | 43.5 | 28.4 | 30.1 | 36.3 |
| | LoRA-One | 0.01 | 28.4 | 38.5 | 51.2 | 45.0 | 30.4 | 31.5 | 37.5 |
| | LoRA-GA | 0.01 | 29.6 | 40.3 | 51.2 | **49.1** | 27.9 | 29.1 | 37.9 |
| LLaMA-2-7B | SpIEL | 0.01 | 30.8 | 40.2 | **53.3** | 43.6 | 33.5 | 30.2 | 38.6 |
| | Fixed Mask | 0.01 | 27.4 | 42.1 | 50.2 | 46.3 | 30.8 | 30.1 | 37.8 |
| | GASDU | 0.01 | **31.4** | **42.7** | 51.7 | 47.8 | **33.5** | **34.4** | **40.2** |
| | Full FT | 100 | 29.6 | 41.8 | 50.0 | 45.5 | 27.7 | 30.3 | 37.5 |
| | LoRA | 0.01 | 60.7 | 47.9 | 65.7 | 56.3 | 64.2 | 45.9 | 56.8 |
| | LoRA-One | 0.01 | 60.5 | 48.6 | 62.3 | 57.7 | 66.1 | 46.5 | 57.0 |
| | LoRA-GA | 0.01 | 61.7 | 50.5 | 70.4 | 60.9 | 61.4 | 48.4 | 58.9 |
| LLaMA-3-8B | SpIEL | 0.01 | 52.6 | 44.7 | 61.3 | 51.7 | 63.9 | 48.2 | 53.7 |
| | Fixed Mask | 0.01 | 63.2 | 51.3 | 63.2 | 58.5 | 65.5 | 47.4 | 58.2 |
| | GASDU | 0.01 | **63.5** | 52.2 | 69.0 | **62.3** | 67.9 | **50.1** | 60.8 |
| | Full FT | 100 | 60.7 | **59.6** | **73.7** | 59.0 | **70.3** | 47.4 | **61.8** |
| | LoRA | 0.01 | 66.9 | 49.5 | 61.3 | 66.3 | 70.1 | 51.8 | 61.0 |
| | LoRA-One | 0.01 | 70.4 | **52.0** | 63.0 | **68.2** | **71.1** | **53.1** | 63.0 |
| | LoRA-GA | 0.01 | **74.3** | 50.7 | 80.5 | 68.1 | 67.2 | 50.6 | 65.2 |
| GPT-OSS-20B | SpIEL | 0.01 | 66.0 | 49.7 | 65.0 | 62.6 | 67.4 | 48.7 | 59.9 |
| | Fixed Mask | 0.01 | 69.4 | 49.8 | 78.0 | 64.5 | 69.3 | 49.8 | 63.5 |
| | GASDU | 0.01 | 73.3 | 51.1 | **81.4** | 66.4 | 69.9 | 52.9 | **65.8** |

Table 1: NumGLUE Arithmetic Reasoning Results (best in **bold**, second-best underlined).

by a one-epoch grid search for each method–task pair (including baselines) over learning rates $\{1 \times 10^{-6}, 5 \times 10^{-6}, 1 \times 10^{-5}, 5 \times 10^{-5}, 1 \times 10^{-4}\}$ and batch sizes $\{4, 8\}$. The best validation configuration is then used for 3-epoch fine-tuning (see Appendix C for per-task setting). For other method-specific hyperparameters, we strictly follow the authors' public implementation. Unless otherwise noted, all runs use DeepSpeed's FusedAdam optimizer (Rasley et al., 2020) and the median score is reported for each method–task pair.

## 5.1 MAIN RESULTS

Tables 1 and 2 report arithmetic (NumGLUE) and commonsense results across the three models. Under a 0.01% update budget, GASDU attains the strongest average among the PEFT baselines on every model. On NumGLUE, GASDU outperforms the baselines on most individual tasks. In terms of averages, it beats the strongest baseline on all backbone models, and it even exceeds full FT on LLaMA-2-7B by roughly 3% (40.2 vs. 37.5) while closely matching full FT on LLaMA-3-8B (60.8 vs. 61.8). On commonsense benchmarks, similar patterns holds: GASDU leads the PEFT baselines and typically comes within a point of full FT on LLaMA-2-7B and LLaMA-3-8B (83.3 vs. 84.5 and 87.9 vs. 88.7, respectively). Overall, these results show that GASDU combines the efficiency of extreme sparsity with the effectiveness of full FT. Matching, and in some cases surpassing, full FT while updating only 0.01% of parameters underscores the promise of gradient-guided dynamic updates for parameter-efficient adaptation on both specialized (NumGLUE) and general (commonsense) reasoning tasks.

## 5.2 EFFECT OF REFRESH PERIOD

Figure 4 demonstrates that GASDU with moderate refresh intervals ($M = 10, 50, 100, 200$) maintains the gradient retention factor $\alpha_t$ around 85% throughout training, ensuring stable preservation of gradient information. When $M$ becomes large (e.g., $M = 400$ or $M = 800$), $\alpha_t$ temporarily declines until the mask is refreshed, and later stages of fine-tuning still recover to the 85% level after updates. This behavior is consistent with the "critical learning regime" emphasized in STEP (Lu et al., 2023) and the dense-to-sparse warm-up used in SparseLoRA (Khaki et al., 2025): early in training, gradient statistics are changing rapidly, so stale masks with large $M$ quickly lose alignment and $\alpha_t$ drops, whereas later, once the gradients stabilize, refreshed masks remain well aligned for many steps and long mask reuse becomes both safe and efficient. In contrast, the Fixed Mask baseline exhibits a steady decline in $\alpha_t$ from about 85% to 65%, indicating progressively weaker gradient signals. Table 3 confirms this behavior in downstream performance: dynamic masks with moderate $M$ consistently outperform the Fixed Mask baseline, with $M = 1$ achieving the best overall average, closely followed by $M = 50$. Larger $M$ values, such as $400$, remain competitive but show slight degradation. These results suggest that choosing a moderate refresh interval effec-

| Model | Method | Upd.% | BoolQ | PIQA | SIQA | HellaSwag | WinoG. | ARC-E | ARC-C | OBQA | Avg |
|-------|--------|-------|-------|------|------|-----------|--------|-------|-------|------|-----|
| | LoRA | 0.01 | 72.0 | 81.4 | 80.5 | 90.1 | 81.2 | 84.3 | 69.2 | 77.4 | 79.5 |
| | LoRA-One | 0.01 | 75.3 | 84.5 | 83.8 | 93.4 | 84.8 | 85.6 | 69.9 | 80.3 | 82.2 |
| | LoRA-GA | 0.01 | 75.9 | 84.9 | 83.5 | 93.6 | 85.6 | 88.1 | 73.8 | 81.0 | 83.0 |
| LLaMA-2-7B | SpIEL | 0.01 | 73.8 | 82.4 | 81.3 | 89.6 | 83.4 | 81.2 | 66.1 | 70.5 | 78.5 |
| | Fixed Mask | 0.01 | 74.0 | 83.7 | 82.8 | 92.2 | 84.4 | 84.5 | 69.4 | 77.6 | 81.1 |
| | GASDU | 0.01 | 76.5 | 85.6 | 85.4 | 94.1 | 86.6 | 86.1 | 70.6 | 81.6 | 83.3 |
| | Full FT | 100 | 77.5 | 86.0 | 83.5 | 93.5 | 91.6 | 87.8 | 72.5 | 83.6 | 84.5 |
| | LoRA | 0.01 | 74.6 | 87.4 | 82.7 | 92.6 | 84.8 | 92.0 | 79.2 | 85.6 | 84.9 |
| | LoRA-One | 0.01 | 77.2 | 90.2 | 85.5 | 95.5 | 87.0 | 94.8 | 82.4 | 88.8 | 87.7 |
| | LoRA-GA | 0.01 | 76.0 | 90.1 | 85.4 | 95.2 | 91.3 | 94.7 | 82.1 | 87.6 | 87.8 |
| LLaMA-3-8B | SpIEL | 0.01 | 71.2 | 87.2 | 84.3 | 92.2 | 83.2 | 92.0 | 76.5 | 83.9 | 83.8 |
| | Fixed Mask | 0.01 | 75.5 | 89.0 | 84.4 | 94.7 | 86.8 | 93.8 | 84.2 | 86.7 | 86.9 |
| | GASDU | 0.01 | 77.2 | 89.6 | 85.8 | 95.3 | 87.0 | 95.2 | 83.5 | 89.9 | 87.9 |
| | Full FT | 100 | 78.0 | 90.5 | 85.1 | 95.3 | 95.0 | 93.8 | 82.5 | 89.5 | 88.7 |
| | LoRA | 0.01 | 73.2 | 88.2 | 82.3 | 92.3 | 82.1 | 94.2 | 89.2 | 90.0 | 86.4 |
| | LoRA-One | 0.01 | 76.8 | 91.2 | 85.2 | 95.3 | 85.8 | 97.1 | 92.5 | 93.3 | 89.6 |
| | LoRA-GA | 0.01 | 76.0 | 90.8 | 84.7 | 94.9 | 88.9 | 96.9 | 92.5 | 93.6 | 89.8 |
| GPT-OSS-20B | SpIEL | 0.01 | 74.6 | 90.6 | 84.3 | 93.1 | 84.8 | 95.9 | 92.2 | 88.0 | 87.9 |
| | Fixed Mask | 0.01 | 73.3 | 89.7 | 83.3 | 93.4 | 86.2 | 96.0 | 91.6 | 90.0 | 87.9 |
| | GASDU | 0.01 | 77.1 | 92.4 | 85.6 | 95.5 | 85.0 | 98.8 | 94.9 | 91.3 | 90.1 |

Table 2: Commonsense Reasoning Results (best in **bold**, second-best underlined).

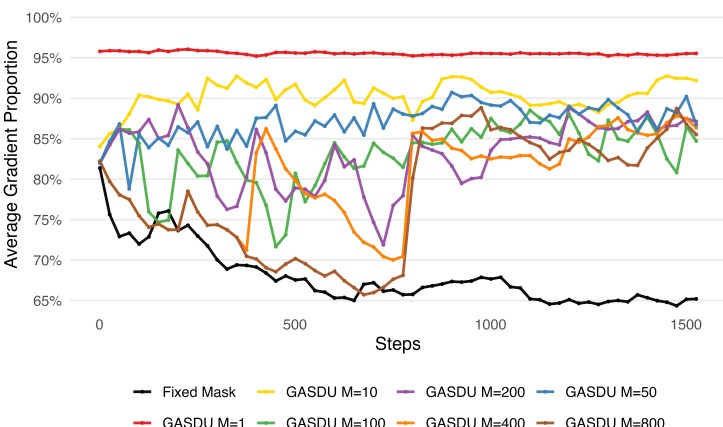

Figure 4: Ratio of masked to full gradient $\ell_2$-norms (smoothed with a 25-step moving average) on the ARC-C dataset for GASDU with various refresh periods ($M$), alongside a Fixed Mask baseline. At every training step $t$, we compute the gradient retention factor $\alpha_t$ (Eq. (4)) in each selected projection layer and report the mean value across all layers.

tively balances computational efficiency and task accuracy, while very small or very large $M$ offer diminishing returns.

| Model | Refresh Period $M$ | Upd. % | type1 | type2 | type3 | type4 | type8 | ARC-C | ARC-E | BoolQ | OBQA | PIQA | Avg |
|-------|--------------------|--------|-------|-------|-------|-------|-------|-------|-------|-------|------|------|-----|
| | 1 | 0.01 | 74.1 | 53.8 | 81.5 | 67.7 | 54.3 | 94.2 | 98.1 | 77.4 | 92.9 | 92.6 | 78.7 |
| | 10 | 0.01 | 72.9 | 50.8 | 80.9 | 66.1 | 52.4 | 93.8 | 98.5 | 76.7 | 92.6 | 92.9 | 77.8 |
| GPT-OSS-20B | 50 | 0.01 | 73.3 | 51.1 | 81.4 | 66.4 | 52.9 | 94.9 | 98.8 | 77.1 | 92.4 | 92.4 | 78.0 |
| | 100 | 0.01 | 69.1 | 51.7 | 82.7 | 66.4 | 52.4 | 93.7 | 98.5 | 76.5 | 93.1 | 92.3 | 77.6 |
| | 400 | 0.01 | 71.6 | 52.9 | 79.6 | 65.9 | 51.2 | 93.4 | 98.4 | 76.1 | 92.2 | 92.3 | 77.4 |
| | Fixed Mask | 0.01 | 69.4 | 49.8 | 78.0 | 64.5 | 49.8 | 91.6 | 96.0 | 73.3 | 90.0 | 89.7 | 75.2 |

Table 3: Results of GASDU on GPT-OSS-20B with different refresh periods $M$ and update percentage (all set to 0.01). Best results are in **bold**, second-best are underlined, on selected NumGLUE arithmetic tasks and commonsense reasoning tasks. Averages are computed over all tasks.

## 5.3 TRAINING EFFICIENCY AND MASK-REFRESH OVERHEAD ANALYSIS

In Table 4a, we compare training efficiency of GASDU ($M$=50) with Full FT, LoRA, Fixed Mask, and SpIEL on ARC-C using LLaMA-2-7B (sequence length 768, batch size 4 per GPU, 2×H100) at a 0.01% update budget. All methods run in `bf16` with the FusedAdam optimizer and CPU offloading for Full FT; results are averaged over 10 runs (50 warm-up, 500 measured iterations). GASDU attains a **10.64×** throughput gain over Full FT and reduces peak memory from 52.2 GB to 15.7 GB (70% less). Its throughput closely matches the Fixed Mask variant, within about 3% (16,140 vs. 16,601 tokens/s), which supports our claim in Section 3 that periodic mask refresh adds negligible overhead when amortized over $M$, while remaining competitive with leading PEFT baselines in both speed and memory.

To quantify refresh overhead, we split the training step with mask refresh into a *Top-k Refresh Block* and a *Base Block* and measure the wall clock time of each block under the same setting in previous training efficiency analysis. The Refresh Block identifies the $k$ largest-magnitude gradient entries by streaming small tiles of the full gradient matrix, merges candidates, and installs the new mask (see implementation details in Section 3); the Base Block performs the standard forward, backward, and masked parameter update. As shown in Table 4b, the Refresh Block time changes only mildly as the update percentage spans several orders of magnitude. The Base Block dominates runtime at high update percentage and is required by all mask-based methods. With a modest refresh period $M$, only one in $M$ steps pays the refresh cost, so the amortized overhead is negligible. Hence, GASDU can achieve throughput on par with static-mask methods across sparsity levels while retaining the benefits of dynamic mask selection.

| LLaMA-2-7B on ARC-C | | |
|---|---|---|
| **Method** | **Throughput (tokens/s)** | **Speedup (×)** | **Peak GPU (GB)** |
| Full FT | 1516.45 | 1.00 | 52.17 |
| LoRA | 17372.06 | 11.46 | 15.73 |
| SpIEL | 16139.60 | 10.64 | 15.77 |
| Fixed Mask | 16601.21 | 10.95 | 15.71 |
| GASDU ($M = 50$) | 16140.20 | 10.64 | 15.74 |

(a) Training efficiency comparison.

| LLaMA-2-7B on ARC-C | | |
|---|---|---|
| **Upd. (%)** | **Top-$k$ Refresh Block (ms)** | **Base Block (ms)** |
| 0.001 | 516.72 | 410.19 |
| 0.01 | 531.37 | 642.54 |
| 0.10 | 558.90 | 2222.84 |
| 1.00 | 694.49 | 17319.04 |

(b) Single-iteration mask-refresh overhead breakdown.

Table 4: Side-by-side summary of GASDU efficiency (left) and mask-refresh overhead profiling (right).

## 6 CONCLUSION

We presented GASDU, an inference-neutral PEFT method that periodically applies Gauss–Southwell–$k$ selection using the current gradient signal, implemented via a streaming, tile-wise Top-$k$ that maintains an $O(k)$ candidate pool and never materializes dense gradients. Under a local Polyak–Łojasiewicz condition, we prove linear convergence with a retention-based rate and derive a lower bound that quantifies the effect of mask reuse across the $M$-step window. A natural extension is an *adaptive* refresh period $M$: use a small $M$ early to track rapidly changing gradients, then increase $M$ later as masks stabilize to further amortize refresh cost. Empirically, across diverse LLMs and benchmarks, GASDU with only $0.01\%$ trainable parameters consistently outperforms strong PEFT baselines and often matches or exceeds full fine-tuning, while achieving up to **10.64×** higher training throughput and about **70%** lower peak memory, with no added inference latency.

**Reproducibility Statement.** We have taken several steps to facilitate reproduction of our results. The GASDU algorithm is specified precisely in Section 3 (Algorithm 1), including the periodic Gauss–Southwell-$k$ rule and the streaming Top-$k$ refresh; implementation choices needed to match our runtimes are described in the "Speed and memory optimizations" paragraph of Section 3 and the efficiency/overhead study in Section 5.3. All theoretical assumptions (e.g., local PL) and guarantees are stated in Section 4, with complete proofs provided in Appendix A.1 and Appendix A.2. Empirical verification of the local PL condition appears in Section 4.3. Datasets, task definitions, and any filtering/exclusions (e.g., NumGLUE types used) are detailed in Section 5 and Appendix B. Training/evaluation protocols, LLMs, selection of baselines, grid-search ranges, and per-task hyperparameters are documented in Section 5 and Appendix C (Tables 8–10); main results and ablations are in Sections 5.1 and 5.2. We provide an anonymous repository (https://anonymous.4open.science/r/GASDU-B86D/) containing source code for GASDU, configuration files, seeds, and scripts to reproduce all tables and figures, along with instructions to fetch the datasets used. Finally, we include example logs and environment files to pin library versions and hardware settings needed to reproduce throughput and memory numbers reported in Section 5.3.

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

**The Use of Large Language Models.** We used large language models (LLMs) as general-purpose assist tools in two limited ways. *(i) Writing polish:* we employed an LLM to improve clarity, grammar, and concision of author-written passages and captions; any suggested text was reviewed, edited, and verified by the authors, and no passages were accepted verbatim without manual revision. *(ii) Retrieval & discovery:* we used an LLM to surface potentially relevant references and related keywords; all citations included in the paper were independently checked by the authors for accuracy and relevance using the original sources. The conceptual contributions, algorithmic design, theoretical results (assumptions, statements, and proofs), experimental protocols, implementations, and analysis are solely by the authors. The LLM was not used to generate data, results, code, or proofs, and it is not an author or contributor.

# A DEFINITIONS AND PROOFS

## A.1 CONVERGENCE ANALYSIS

**Definition A.1** (*L*-smoothness). A differentiable function $f : \mathbb{R}^d \to \mathbb{R}$ is *L-smooth* on a set $\mathcal{S} \subseteq \mathbb{R}^d$ if its gradient is Lipschitz continuous with constant $L > 0$ over $\mathcal{S}$; that is,

$$\|\nabla f(x) - \nabla f(y)\|_2 \ \leq \ L \, \|x - y\|_2, \qquad \forall \, x, y \in \mathcal{S}.$$

Equivalently, for all $x, y \in \mathcal{S}$,

$$f(y) \ \leq \ f(x) + \langle \nabla f(x), \, y - x \rangle + \frac{L}{2} \, \|y - x\|_2^2.$$

**Theorem 4.2** (Local PL Convergence of GASDU). *Let $f : \mathbb{R}^d \to \mathbb{R}$ be L-smooth and $\mu$-PL on a set $\mathcal{S} \subseteq \mathbb{R}^d$. Assume $W^{(0)} \in \mathcal{S}$ and that the iterations produced by the above update rule remain in $\mathcal{S}$. Then for any stepsize $\gamma \leq 1/L$ the sequence $\{W^{(t)}\}$ satisfies*

$$f\big(W^{(t+1)}\big) - f(W^*) \ \leq \ \big(1 - \alpha_t \, \mu \, \gamma\big) \big[f(W^{(t)}) - f(W^*)\big],$$

*and consequently, if $\alpha = \inf_t \alpha_t > 0$,*

$$f\big(W^{(t)}\big) - f(W^*) \ \leq \ \big(1 - \alpha \, \mu \, \gamma\big)^t \big[f(W^{(0)}) - f(W^*)\big].$$

*Proof of Theorem 4.2.* Let $G^{(t)} := \nabla f(W^{(t)})$. By *L*-smoothness (with $\langle A, B \rangle := \operatorname{tr}(A^\top B)$ the Frobenius inner product and $\|\cdot\|$ its induced norm),

$$f\big(W^{(t+1)}\big) \ \leq \ f\big(W^{(t)}\big) \ + \ \big\langle G^{(t)}, W^{(t+1)} - W^{(t)} \big\rangle \ + \ \frac{L}{2} \, \big\|W^{(t+1)} - W^{(t)}\big\|^2.$$

Using the masked update $W^{(t+1)} - W^{(t)} = -\gamma \big(\Lambda^{(t)} \odot G^{(t)}\big)$, we obtain

$$f\big(W^{(t+1)}\big) \ \leq \ f\big(W^{(t)}\big) \ - \ \gamma \big\langle G^{(t)}, \Lambda^{(t)} \odot G^{(t)} \big\rangle \ + \ \frac{L\gamma^2}{2} \, \big\|\Lambda^{(t)} \odot G^{(t)}\big\|^2.$$

Since $\big\langle G^{(t)}, \Lambda^{(t)} \odot G^{(t)} \big\rangle = \big\|\Lambda^{(t)} \odot G^{(t)}\big\|^2$, it follows that

$$f\big(W^{(t+1)}\big) \ \leq \ f\big(W^{(t)}\big) \ - \ \gamma \Big(1 - \frac{L\gamma}{2}\Big) \big\|\Lambda^{(t)} \odot G^{(t)}\big\|^2.$$

For any $\gamma \leq 1/L$ the factor in parentheses is at least $1/2$, so

$$f\big(W^{(t+1)}\big) \ \leq \ f\big(W^{(t)}\big) \ - \ \frac{\gamma}{2} \big\|\Lambda^{(t)} \odot G^{(t)}\big\|^2. \tag{A.1}$$

By the local $\mu$-PL condition, $\|G^{(t)}\|^2 \geq 2\mu \big[f(W^{(t)}) - f(W^*)\big]$. By definition of $\alpha_t$, $\big\|\Lambda^{(t)} \odot G^{(t)}\big\|^2 = \alpha_t \, \|G^{(t)}\|^2$. Substituting into (A.1) gives

$$f\big(W^{(t+1)}\big) - f(W^*) \ \leq \ \big(1 - \alpha_t \mu \gamma\big) \big[f(W^{(t)}) - f(W^*)\big],$$

and the geometric rate follows by recursion. $\qquad\qquad\square$

## A.2  MASK-REUSE RETENTION ANALYSIS

Between refreshes the update is masked: $W^{(s+1)} = W^{(s)} - \gamma \Lambda g^{(s)}$ with $g^{(s)} := \nabla f(W^{(s)})$ and fixed $\Lambda = \Lambda^{(t_{\mathrm{ref}})}$. Let $b := \|g^{(t_{\mathrm{ref}})}\|_2$ and $a := \|\Lambda g^{(t_{\mathrm{ref}})}\|_2 = \sqrt{\alpha_{t_{\mathrm{ref}}}}\, b$.

**Step 1: Gradient drift.**  By $L$-smoothness,

$$\|g^{(t)} - g^{(t_{\mathrm{ref}})}\| \;\le\; L\,\|W^{(t)} - W^{(t_{\mathrm{ref}})}\| \;\le\; L\gamma \sum_{s=t_{\mathrm{ref}}}^{t-1} \|\Lambda g^{(s)}\| \;\le\; L\gamma\,\sqrt{\tau_t}\Big( \sum_{s=t_{\mathrm{ref}}}^{t-1} \|\Lambda g^{(s)}\|^2 \Big)^{1/2},$$

where the last step uses Cauchy–Schwarz and $\tau_t := t - t_{\mathrm{ref}}$.

**Step 2: Bounding the masked-gradient energy.**  $L$-smoothness and the masked update give the standard descent estimate

$$f(W^{(s+1)}) \le f(W^{(s)}) - \gamma\Big(1 - \tfrac{L\gamma}{2}\Big)\|\Lambda g^{(s)}\|^2.$$

Summing from $s = t_{\mathrm{ref}}$ to $t - 1$ and using $f(W^{(t)}) \ge f(W^*)$,

$$\sum_{s=t_{\mathrm{ref}}}^{t-1} \|\Lambda g^{(s)}\|^2 \;\le\; \frac{f(W^{(t_{\mathrm{ref}})}) - f(W^*)}{\gamma\,(1 - \tfrac{L\gamma}{2})}.$$

By the local $\mu$-PL condition at $W^{(t_{\mathrm{ref}})}$, $b^2 = \|g^{(t_{\mathrm{ref}})}\|^2 \ge 2\mu\,[f(W^{(t_{\mathrm{ref}})}) - f(W^*)]$, hence

$$\sum_{s=t_{\mathrm{ref}}}^{t-1} \|\Lambda g^{(s)}\|^2 \;\le\; \frac{b^2}{2\mu\,\gamma\,(1 - \tfrac{L\gamma}{2})}.$$

**Step 3: Explicit drift bound.**  Combining Steps 1–2 yields

$$\Delta_t := \|g^{(t)} - g^{(t_{\mathrm{ref}})}\| \;\le\; \frac{L}{\sqrt{2\mu}}\,\sqrt{\frac{\gamma\,\tau_t}{1 - \tfrac{L\gamma}{2}}}\; b \;=:\; \rho_t\, b,$$

which proves the stated form of $\rho_t$, and since $\gamma \le 1/L$ implies $(1 - \tfrac{L\gamma}{2})^{-1} \le 2$, also $\rho_t \le \frac{L}{\sqrt{\mu}}\sqrt{\gamma\,\tau_t}$.

**Step 4: Retention under reuse.**  By the triangle inequality,

$$\|\Lambda g^{(t)}\| \;\ge\; \|\Lambda g^{(t_{\mathrm{ref}})}\| - \|g^{(t)} - g^{(t_{\mathrm{ref}})}\| \;\ge\; (a - \Delta_t) \;=\; \big(\sqrt{\alpha_{t_{\mathrm{ref}}}} - \rho_t\big)_+ b,$$

and also $\|g^{(t)}\| \le b + \Delta_t = (1 + \rho_t)b$. Therefore

$$\alpha_t \;=\; \frac{\|\Lambda g^{(t)}\|^2}{\|g^{(t)}\|^2} \;\ge\; \frac{\big[\sqrt{\alpha_{t_{\mathrm{ref}}}} - \rho_t\big]_+^2}{(1 + \rho_t)^2},$$

which is the claimed lower bound.  □

## A.3  ADDITIONAL THEORETICAL ANALYSIS

**Theorem A.1** (Bias–variance for GASDU in the linearized regime)**.** *Assume the linearized data model*

$$y = f_{W^{(0)}}(x) + \phi(x)^\top \theta^\star + \varepsilon, \qquad \varepsilon \sim \mathcal{N}(0, \sigma^2).$$

*Let $\widehat{f}_T$ be the predictor after $T$ iterations of , and suppose (within the linearized model) it attains the least-squares solution in the subspace spanned by $\mathcal{S}_T := \bigcup_{t=0}^{T-1} \mathrm{supp}(\Lambda^{(t)})$, with $s_T = |\mathcal{S}_T|$. If $\|\phi(x)\|_2 \le B$ for test $x$, then*

$$\Big[\big(\widehat{f}_T(x) - f_{W^{(0)}}(x) - \phi(x)^\top \theta^\star\big)^2\Big] = \underbrace{\|\Pi_{\mathcal{S}_T^\perp} \Phi\theta^\star\|_{L_2}^2}_{\mathrm{bias}^2} + \underbrace{\sigma^2\,\mathrm{tr}\Big((\Phi_{\mathcal{S}_T}^\top \Phi_{\mathcal{S}_T})^{-1}\Sigma_{\mathcal{S}_T}\Big)/n}_{\mathrm{variance}},$$

*where $\Sigma_{\mathcal{S}_T} = [\phi_{\mathcal{S}_T}(x)\phi_{\mathcal{S}_T}(x)^\top]$. In particular, if $\lambda_{\min}(\Phi_{\mathcal{S}_T}^\top \Phi_{\mathcal{S}_T}/n) \ge \lambda_0 > 0$ and $\|\phi(x)\|_2 \le B$, then*

$$\Big[\big(\widehat{f}_T(x) - f_{W^{(0)}}(x) - \phi(x)^\top \theta^\star\big)^2\Big] \;\le\; \|\Pi_{\mathcal{S}_T^\perp} \Phi\theta^\star\|_{L_2}^2 \;+\; \sigma^2\,\frac{B^2\,s_T}{\lambda_0\,n}.$$

*Proof.* Standard linear-regression bias–variance on the feature-restricted design $\Phi_{\mathcal{S}_T}$. The inequality uses $\operatorname{tr}(A^{-1}B) \leq \lambda_{\min}(A)^{-1}\operatorname{tr}(B)$ and $\operatorname{tr}(\Sigma_{\mathcal{S}_T}) \leq B^2 s_T$. $\qquad\square$

**Theorem A.2** (Uniform stability of under PL with retention). *Let $(W; z)$ be -smooth and -Lipschitz in $W$, and suppose the empirical loss $f(W) = \frac{1}{n}\sum_{i=1}^{n}(W; z_i)$ satisfies a local Polyak–Łojasiewicz (PL) inequality $\frac{1}{2}\|\nabla f(W)\|_2^2 \geq \big(f(W) - f^\star\big)$ on a set containing all iterates. Run with steps $\gamma_t \leq 1/$ and masks $\Lambda^{(t)}$, and define*

$$\alpha_t := \frac{\|\Lambda^{(t)} \odot \nabla f(W^{(t)})\|_2^2}{\|\nabla f(W^{(t)})\|_2^2} \in [0, 1].$$

*Then is $\epsilon$-uniformly stable with*

$$\epsilon \leq \frac{2^2}{n} \sum_{t=0}^{T-1} \gamma_t \prod_{s=t+1}^{T-1} \big(1 - \alpha_s \, \gamma_s\big).$$

*In particular, for constant $\gamma \leq 1/$ and $\alpha = \inf_t \alpha_t > 0$,*

$$\epsilon \leq \frac{2^2}{n} \cdot \frac{1 - (1 - \alpha\gamma)^T}{\alpha}.$$

*Proof.* Couple two runs on neighboring datasets and apply the Hardt–Recht–Singer SGD-stability recursion (Hardt et al., 2016). PL with masked descent yields a per-step contraction factor $(1 - \alpha_t\gamma_t)$ in function value (cf. Karimi et al., 2016). Unroll and sum the sensitivities as in (Hardt et al., 2016). $\qquad\square$

**Theorem A.3** (Complexity of sparse linearized hypotheses). *Let $\mathcal{H}_{s,R} = \{x \mapsto f_{W^{(0)}}(x) + \phi(x)^\top\theta : \|\theta\|_0 \leq s, \|\theta\|_2 \leq R\}$, and assume $\|\phi(x)\|_2 \leq B$. Then*

$$\mathfrak{R}_n(\mathcal{H}_{s,R}) \leq \frac{BR}{\sqrt{n}}\sqrt{2s \log \frac{ed}{s}}.$$

*Hence, for -Lipschitz losses, the expected generalization gap is $O\Big(\frac{BR}{\sqrt{n}}\sqrt{s \log(ed/s)}\Big)$.*

*Proof.* Apply standard sparse linear class bounds via Maurey sparsification / covering arguments (e.g., Bartlett & Mendelson, 2002). The baseline $f_{W^{(0)}}$ is fixed and does not affect complexity. $\qquad\square$

**Theorem A.4** (Compressible gradients preserve energy). *Let $g \in \mathbb{R}^d$ with nonincreasing magnitudes $|g|_{(1)} \geq \cdots \geq |g|_{(d)}$, and suppose $g \in \ell_{p,\infty}$ with $\|g\|_{p,\infty} := \sup_{i\geq 1} i^{1/p}|g|_{(i)} \leq C$ for some $p \in (0, 2)$. Let $g_{1:k}$ be the Top-$k$ truncation. Then*

$$\frac{\|g_{1:k}\|_2^2}{\|g\|_2^2} \geq 1 - c_p \frac{C^2}{\|g\|_2^2} k^{1-\frac{2}{p}},$$

*with $c_p = \frac{p}{2-p}$. Consequently, if $\Lambda$ keeps the Top-$k$, the retention $\alpha = \|\Lambda \odot g\|_2^2/\|g\|_2^2$ satisfies $\alpha \geq 1 - c_p\,(C^2/\|g\|_2^2)\,k^{1-2/p}$.*

*Proof.* Tail bound for weak-$\ell_p$: $|g|_{(i)} \leq C\,i^{-1/p}$ gives $\sum_{i>k}|g|_{(i)}^2 \leq C^2\sum_{i>k}i^{-2/p} \leq \frac{p}{2-p}C^2 k^{1-2/p}$ (see, e.g., Foucart & Rauhut, 2013, Section 1.3). Conclude by decomposing $\|g\|_2^2$ into head+tail. $\qquad\square$

**Theorem A.5** (Top-$k$ recovery under sub-Gaussian perturbations). *Let $g \in \mathbb{R}^d$ be the true gradient and $\widehat{g} = g + \xi$, where $\xi$ has independent mean-zero $\sigma^2$-sub-Gaussian coordinates. Let $\mathcal{T}_k(g)$ be the Top-$k$ index set and $\Delta_k := |g|_{(k)} - |g|_{(k+1)} > 0$. If*

$$\Delta_k \geq 2\sigma\sqrt{2\log(d/\delta)},$$

*then with probability at least $1 - \delta$, $\mathcal{T}_k(\widehat{g}) = \mathcal{T}_k(g)$ and thus $\alpha(\widehat{g}) = \alpha(g)$. Generally, $[\alpha(\widehat{g})] \geq \alpha(g) - \Pr(\mathcal{T}_k(\widehat{g}) \neq \mathcal{T}_k(g))$.*

*Proof.* Control order-statistic flips by a union bound and sub-Gaussian tails; see, e.g., Vershynin (2018, Ch. 2). If no pairwise swap occurs across the $k$-th threshold, the two Top-$k$ sets coincide. $\square$

**Theorem A.6** (Support recovery under mutual coherence). *In the linearized model with design $\Phi \in \mathbb{R}^{n \times d}$, suppose $\theta^\star$ is $s$-sparse and the mutual coherence $\mu(\Phi) := \max_{i \neq j} \frac{|\Phi_{\cdot i} \Phi_{\cdot j}|}{\|\Phi_{\cdot i}\|_2 \|\Phi_{\cdot j}\|_2}$ satisfies $\mu(\Phi) < \frac{1}{2s-1}$. Run with periodic refreshes that add $k$ new coordinates by selecting the $k$ largest correlations with the residual (equivalently, Top-$k$ gradient magnitudes in the LS subproblem), and perform least squares on the active set between refreshes. Then after $R \geq \lceil s/k \rceil$ refreshes, the active set contains $\mathrm{supp}(\theta^\star)$. With sub-Gaussian noise of variance $\sigma^2$, the LS estimator over the recovered support satisfies*

$$\|\widehat{\theta} - \theta^\star\|_2 \lesssim \sigma \sqrt{\frac{s \log d}{n}},$$

*with constants depending only on $\mu(\Phi)$.*

*Proof.* Batch-$k$ variant of standard OMP/Stagewise analyses (e.g., Tropp & Gilbert, 2007; Nutini et al., 2015): coherence ensures each refresh includes a true index; after $\lceil s/k \rceil$ refreshes the true support is included. Noisy rates follow from restricted eigenvalue/coherence arguments. $\square$

**Theorem A.7** (Excess risk via retention and sparsity). *Under the assumptions of Theorems A.1 and A.3, run with stepsizes $\gamma_t \leq 1/$ for $T$ steps and define $\underline{\alpha} := \inf_{0 \leq t < T} \alpha_t > 0$. Let $R_T := \sum_{t=0}^{T-1} \gamma_t \|\Lambda^{(t)} \odot \nabla f(W^{(t)})\|_2$. Then for any $\delta \in (0, 1)$, with probability at least $1 - \delta$,*

$$\mathcal{E}(\widehat{f}_T) := [(\widehat{f}_T)] - [(f^\star)] \lesssim \underbrace{\|\Pi_{\mathcal{S}_T^\perp} \Phi \theta^\star\|_{L_2}^2}_{bias} + \underbrace{\sigma^2 \frac{s_T}{n}}_{variance} + \underbrace{\frac{B R_T}{\sqrt{n}} \sqrt{s_T \log(ed/s_T)}}_{estimation} + \underbrace{(1 - \underline{\alpha} \, \bar{\gamma})^T \Delta_0}_{opt. \; error},$$

*where $\bar{\gamma} = \min_t \gamma_t$ and $\Delta_0 = f(W^{(0)}) - f(W^\star)$.*

*Proof.* Combine Theorems A.1 and A.3 to control approximation and estimation terms (see also Bartlett & Mendelson, 2002). Optimization error follows from PL with retained energy: $f(W^{(t+1)}) - f^\star \leq (1 - \alpha_t \gamma_t)(f(W^{(t)}) - f^\star)$ (**?**). $\square$

[Within-window retention decay] Let $t_0$ be a refresh step where the mask is fixed for $t = t_0, \ldots, t_0 + M - 1$, and suppose $f$ is -smooth. Define $\alpha_t = \|\Lambda^{(t_0)} \odot \nabla f(W^{(t)})\|_2^2 / \|\nabla f(W^{(t)})\|_2^2$ for $t_0 \leq t < t_0 + M$. Then for any $t$ in this window,

$$\alpha_t \geq \alpha_{t_0} - 2 \sum_{s=t_0}^{t-1} \frac{\|\Lambda^{(t_0)} \odot (\nabla f(W^{(s+1)}) - \nabla f(W^{(s)}))\|_2}{\|\nabla f(W^{(t)})\|_2}.$$

In particular, with updates and $\gamma_s \leq \gamma$,

$$\alpha_t \geq \alpha_{t_0} - 2\gamma \sum_{s=t_0}^{t-1} \|\Lambda^{(s)} \odot \nabla f(W^{(s)})\|_2 \geq \alpha_{t_0} - O(M\gamma) \cdot \max_s \|\Lambda^{(s)} \odot \nabla f(W^{(s)})\|_2.$$

*Proof.* Let $P = \mathrm{diag}(\Lambda^{(t_0)})$ and $g_t = \nabla f(W^{(t)})$. Then $\|Pg_t\|_2^2 - \|Pg_{t_0}\|_2^2 = \sum_{s=t_0}^{t-1} 2Pg_s P(g_{s+1} - g_s) + \|P(g_{s+1} - g_s)\|_2^2$. Drop the nonnegative quadratic term and apply Cauchy–Schwarz. Smoothness gives $\|g_{s+1} - g_s\|_2 \leq \|W^{(s+1)} - W^{(s)}\|_2 = \gamma_s \|\Lambda^{(s)} \odot g_s\|_2$, completing the bound. $\square$

# B   DATASET DESCRIPTIONS

**NumGLUE Sub-datasets.**   The NumGLUE benchmark consists of eight arithmetic reasoning tasks, four of which are newly curated and four adapted from existing datasets (Mishra et al., 2022). Type 1 (*Commonsense + Arithmetic Reasoning*) combines everyday numerical facts with simple calculations. For example, it requires knowing that a human has two hands and multiplying this fact by the number of people in a scenario. Type 2 (*Domain-specific + Arithmetic Reasoning*) requires scientific knowledge, such as chemical reaction stoichiometry or physical laws, combined with arithmetic operations. Type3 (*Commonsense + Quantitative Comparison*) asks models to compare quantities in everyday contexts, such as which object experiences greater gravitational force given their respective masses. Type 4 (*Fill-in-the-blank Arithmetic*) presents arithmetic word problems reformatted into completion-style questions, where a key value must be inferred and filled in. Type 5 (*Reading Comprehension with Explicit Numerical Reasoning*) is drawn from DROP (Dua et al., 2019), where the answer must be a number following arithmetic operations over textual spans. Type 6 (*Reading Comprehension with Implicit Numerical Reasoning*) also originates from DROP, but its answers are textual entities rather than numbers. Type 7 (*Quantitative Natural Language Inference*) comes from EQUATE (Ravichander et al., 2019), where the task is to classify a premise–hypothesis pair as entailment, contradiction, or neutral based on numerical reasoning. Finally, Type 8 (*Arithmetic Word Problems*) collects classic math problems from sources such as MAWPS (Koncel-Kedziorski et al., 2016) and earlier algebra problem datasets (Kushman et al., 2014), focusing squarely on direct arithmetic manipulation. We train and evaluate on the official training and test splits released by the authors of NumGLUE.

In our evaluation, we exclude Type 6 and Type 7. Both tasks differ from the others in that they do not require numeric outputs. Type 6 expects entity-level answers extracted from passages, while Type 7 is a natural language inference classification task. Since our study emphasizes arithmetic reasoning with explicit numeric predictions, we restrict our NumGLUE subset to Types 1–5 and Type 8, which directly measure numerical accuracy.

**Commonsense Reasoning Benchmarks.**   Beyond arithmetic reasoning, we also evaluate on a suite of established commonsense reasoning datasets. BoolQ (Clark et al., 2019) tests yes/no question answering against passages. PIQA (Bisk et al., 2020) centers on physical commonsense by asking models to choose the more plausible of two candidate actions. SocialIQA (Sap et al., 2019) probes understanding of social interactions and motivations. HellaSwag (Zellers et al., 2019) presents adversarially filtered sentence completion problems. Winogrande (Sakaguchi et al., 2020) is a large-scale coreference resolution benchmark requiring commonsense disambiguation. ARC-Easy and ARC-Challenge (Clark et al., 2018) are multiple-choice science exams of varying difficulty. OpenBookQA (Mihaylov et al., 2018) blends scientific knowledge with commonsense inference.

Together, these sub-datasets provide a comprehensive test bed. NumGLUE targets fine-grained arithmetic reasoning skills, while the commonsense suite evaluates broader physical, social, and scientific inference abilities. This combination allows us to assess whether parameter-efficient fine-tuning methods such as GASDU can adapt large language models to diverse reasoning domains.

## C  BATCH SIZE AND LEARNING RATE

| Method | Task | LR | Batch |
|---|---|---|---|
| SpIEL | BoolQ | 1e-05 | 4 |
| | PIQA | 1e-05 | 4 |
| | SocialIQA | 5e-05 | 4 |
| | HellaSwag | 1e-05 | 4 |
| | WinoGrande | 1e-05 | 4 |
| | ARC-Easy | 1e-05 | 4 |
| | ARC-Challenge | 5e-05 | 4 |
| | OBQA | 5e-05 | 4 |
| Full | BoolQ | 1e-05 | 4 |
| | PIQA | 1e-05 | 4 |
| | SocialIQA | 1e-05 | 4 |
| | HellaSwag | 1e-04 | 4 |
| | WinoGrande | 1e-05 | 4 |
| | ARC-Easy | 1e-05 | 4 |
| | ARC-Challenge | 1e-05 | 4 |
| | OBQA | 1e-05 | 4 |
| LoRA | BoolQ | 1e-04 | 4 |
| | PIQA | 1e-04 | 4 |
| | SocialIQA | 1e-04 | 4 |
| | HellaSwag | 1e-04 | 4 |
| | WinoGrande | 1e-04 | 4 |
| | ARC-Easy | 1e-04 | 4 |
| | ARC-Challenge | 1e-04 | 4 |
| | OBQA | 1e-04 | 4 |
| Fixed Mask | BoolQ | 5e-05 | 4 |
| | PIQA | 1e-04 | 8 |
| | SocialIQA | 1e-04 | 4 |
| | HellaSwag | 1e-04 | 8 |
| | WinoGrande | 1e-04 | 4 |
| | ARC-Easy | 1e-04 | 4 |
| | ARC-Challenge | 1e-04 | 4 |
| | OBQA | 1e-04 | 4 |
| GASDU | BoolQ | 1e-04 | 8 |
| | PIQA | 1e-04 | 8 |
| | SocialIQA | 1e-04 | 8 |
| | HellaSwag | 1e-04 | 4 |
| | WinoGrande | 1e-04 | 8 |
| | ARC-Easy | 1e-04 | 8 |
| | ARC-Challenge | 1e-04 | 4 |
| | OBQA | 1e-04 | 8 |

Table 5: Training hyperparameters for **LLaMA-2-7B** of Commonsense Reasoning dataset.

| Method | Task | LR | Batch |
|---|---|---|---|
| SpIEL | BoolQ | 5e-06 | 4 |
| | PIQA | 5e-06 | 4 |
| | SocialIQA | 5e-06 | 4 |
| | HellaSwag | 1e-06 | 4 |
| | WinoGrande | 5e-06 | 4 |
| | ARC-Easy | 1e-06 | 4 |
| | ARC-Challenge | 1e-05 | 4 |
| | OBQA | 5e-06 | 4 |
| Full | BoolQ | 5e-06 | 4 |
| | PIQA | 5e-06 | 4 |
| | SocialIQA | 5e-06 | 4 |
| | HellaSwag | 5e-06 | 8 |
| | WinoGrande | 1e-05 | 4 |
| | ARC-Easy | 5e-06 | 4 |
| | ARC-Challenge | 5e-06 | 4 |
| | OBQA | 5e-06 | 4 |
| LoRA | BoolQ | 1e-04 | 4 |
| | PIQA | 1e-04 | 4 |
| | SocialIQA | 1e-04 | 8 |
| | HellaSwag | 1e-04 | 8 |
| | WinoGrande | 1e-04 | 8 |
| | ARC-Easy | 1e-04 | 4 |
| | ARC-Challenge | 1e-04 | 4 |
| | OBQA | 1e-04 | 8 |
| Fixed Mask | BoolQ | 5e-05 | 8 |
| | PIQA | 5e-05 | 4 |
| | SocialIQA | 1e-04 | 4 |
| | HellaSwag | 1e-04 | 8 |
| | WinoGrande | 1e-04 | 8 |
| | ARC-Easy | 5e-05 | 8 |
| | ARC-Challenge | 5e-05 | 4 |
| | OBQA | 1e-04 | 8 |
| GASDU | BoolQ | 5e-05 | 8 |
| | PIQA | 5e-05 | 8 |
| | SocialIQA | 5e-05 | 8 |
| | HellaSwag | 5e-05 | 8 |
| | WinoGrande | 5e-05 | 8 |
| | ARC-Easy | 5e-05 | 4 |
| | ARC-Challenge | 5e-05 | 4 |
| | OBQA | 5e-05 | 8 |

Table 6: Training hyperparameters for **LLaMA-3-8B** of Commonsense Reasoning dataset.

| Method | Task | LR | Batch |
|--------|------|-----|-------|
| SpIEL | BoolQ | 5e-06 | 4 |
| | PIQA | 1e-06 | 4 |
| | SocialIQA | 1e-06 | 4 |
| | HellaSwag | 1e-06 | 4 |
| | WinoGrande | 5e-06 | 8 |
| | ARC-Easy | 5e-06 | 4 |
| | ARC-Challenge | 5e-06 | 8 |
| | OBQA | 1e-05 | 8 |
| LoRA | BoolQ | 1e-04 | 8 |
| | PIQA | 1e-04 | 4 |
| | SocialIQA | 1e-04 | 4 |
| | HellaSwag | 1e-04 | 8 |
| | WinoGrande | 1e-04 | 4 |
| | ARC-Easy | 5e-05 | 8 |
| | ARC-Challenge | 5e-05 | 4 |
| | OBQA | 1e-04 | 4 |
| Fixed Mask | BoolQ | 1e-05 | 8 |
| | PIQA | 1e-05 | 8 |
| | SocialIQA | 1e-05 | 4 |
| | HellaSwag | 1e-05 | 4 |
| | WinoGrande | 5e-05 | 8 |
| | ARC-Easy | 1e-05 | 8 |
| | ARC-Challenge | 5e-06 | 4 |
| | OBQA | 1e-05 | 4 |
| GASDU | BoolQ | 1e-05 | 8 |
| | PIQA | 5e-06 | 8 |
| | SocialIQA | 1e-05 | 4 |
| | HellaSwag | 1e-05 | 8 |
| | WinoGrande | 1e-05 | 8 |
| | ARC-Easy | 1e-05 | 4 |
| | ARC-Challenge | 1e-05 | 4 |
| | OBQA | 5e-06 | 8 |

Table 7: Training hyperparameters for **GPT-OSS-20B** of Commonsense Reasoning dataset.

| Method | Task | LR | Batch |
|---|---|---|---|
| SpIEL | Type1 | 1e-06 | 8 |
| | Type2 | 1e-05 | 4 |
| | Type3 | 1e-05 | 4 |
| | Type4 | 5e-05 | 8 |
| | Type5 | 5e-06 | 8 |
| | Type8 | 1e-05 | 4 |
| Full | Type1 | 5e-06 | 4 |
| | Type2 | 1e-05 | 4 |
| | Type3 | 1e-06 | 4 |
| | Type4 | 1e-05 | 4 |
| | Type5 | 1e-06 | 4 |
| | Type8 | 5e-06 | 4 |
| LoRA | Type1 | 5e-05 | 4 |
| | Type2 | 5e-05 | 4 |
| | Type3 | 1e-05 | 8 |
| | Type4 | 1e-04 | 4 |
| | Type5 | 5e-05 | 4 |
| | Type8 | 1e-04 | 8 |
| LoRA-One | Type1 | 5e-05 | 4 |
| | Type2 | 5e-05 | 8 |
| | Type3 | 1e-06 | 4 |
| | Type4 | 1e-04 | 4 |
| | Type5 | 5e-05 | 4 |
| | Type8 | 5e-05 | 4 |
| LoRA-GA | Type1 | 1e-04 | 4 |
| | Type2 | 5e-05 | 4 |
| | Type3 | 1e-06 | 8 |
| | Type4 | 5e-05 | 4 |
| | Type5 | 5e-6 | 4 |
| | Type8 | 5e-05 | 4 |
| Fixed Mask | Type1 | 5e-05 | 4 |
| | Type2 | 1e-04 | 4 |
| | Type3 | 1e-06 | 4 |
| | Type4 | 1e-04 | 8 |
| | Type5 | 1e-05 | 8 |
| | Type8 | 5e-05 | 8 |
| GASDU | Type1 | 5e-05 | 4 |
| | Type2 | 5e-05 | 8 |
| | Type3 | 1e-06 | 8 |
| | Type4 | 5e-05 | 4 |
| | Type5 | 1e-05 | 4 |
| | Type8 | 5e-05 | 8 |

Table 8: Training hyperparameters for **LLaMA-2-7B** on the NumGLUE dataset.

| Method | Task | LR | Batch |
|---|---|---|---|
| SpIEL | Type1 | 5e-06 | 4 |
|  | Type2 | 5e-05 | 4 |
|  | Type3 | 1e-05 | 4 |
|  | Type4 | 1e-04 | 8 |
|  | Type5 | 5e-06 | 8 |
|  | Type8 | 5e-06 | 4 |
| Full | Type1 | 5e-06 | 4 |
|  | Type2 | 5e-06 | 4 |
|  | Type3 | 5e-06 | 4 |
|  | Type4 | 5e-06 | 4 |
|  | Type5 | 5e-06 | 4 |
|  | Type8 | 1e-06 | 4 |
| LoRA | Type1 | 1e-04 | 4 |
|  | Type2 | 1e-04 | 4 |
|  | Type3 | 1e-04 | 4 |
|  | Type4 | 1e-04 | 8 |
|  | Type5 | 1e-04 | 8 |
|  | Type8 | 1e-04 | 4 |
| LoRA-One | Type1 | 1e-04 | 4 |
|  | Type2 | 1e-04 | 4 |
|  | Type3 | 5e-05 | 4 |
|  | Type4 | 1e-04 | 4 |
|  | Type5 | 1e-04 | 8 |
|  | Type8 | 1e-04 | 8 |
| LoRA-GA | Type1 | 1e-04 | 4 |
|  | Type2 | 1e-04 | 8 |
|  | Type3 | 1e-04 | 4 |
|  | Type4 | 1e-04 | 4 |
|  | Type5 | 1e-05 | 4 |
|  | Type8 | 1e-05 | 4 |
| Fixed Mask | Type1 | 5e-05 | 4 |
|  | Type2 | 5e-05 | 8 |
|  | Type3 | 5e-05 | 4 |
|  | Type4 | 1e-04 | 8 |
|  | Type5 | 5e-05 | 8 |
|  | Type8 | 5e-05 | 8 |
| GASDU | Type1 | 1e-04 | 4 |
|  | Type2 | 5e-05 | 8 |
|  | Type3 | 5e-05 | 4 |
|  | Type4 | 1e-04 | 8 |
|  | Type5 | 5e-05 | 8 |
|  | Type8 | 5e-05 | 8 |

Table 9: Training hyperparameters for **LLaMA-3-8B** on the NumGLUE dataset.

| Method | Task | LR | Batch |
|--------|------|------|-------|
| SpIEL | Type1 | 1e-06 | 8 |
| | Type2 | 5e-06 | 8 |
| | Type3 | 1e-05 | 8 |
| | Type4 | 1e-06 | 4 |
| | Type5 | 1e-06 | 8 |
| | Type8 | 1e-05 | 8 |
| LoRA | Type1 | 1e-04 | 4 |
| | Type2 | 1e-04 | 4 |
| | Type3 | 5e-05 | 8 |
| | Type4 | 1e-04 | 4 |
| | Type5 | 5e-05 | 8 |
| | Type8 | 1e-04 | 4 |
| LoRA-One | Type1 | 1e-04 | 4 |
| | Type2 | 1e-04 | 4 |
| | Type3 | 5e-05 | 8 |
| | Type4 | 1e-04 | 4 |
| | Type5 | 5e-05 | 4 |
| | Type8 | 1e-04 | 4 |
| LoRA-GA | Type1 | 1e-04 | 4 |
| | Type2 | 1e-04 | 4 |
| | Type3 | 1e-04 | 4 |
| | Type4 | 1e-04 | 4 |
| | Type5 | 5e-06 | 4 |
| | Type8 | 5e-05 | 4 |
| Fixed Mask | Type1 | 1e-05 | 4 |
| | Type2 | 1e-05 | 4 |
| | Type3 | 1e-04 | 8 |
| | Type4 | 1e-05 | 8 |
| | Type5 | 1e-05 | 8 |
| | Type8 | 1e-05 | 8 |
| GASDU | Type1 | 1e-05 | 4 |
| | Type2 | 5e-06 | 4 |
| | Type3 | 5e-05 | 4 |
| | Type4 | 1e-05 | 8 |
| | Type5 | 5e-06 | 8 |
| | Type8 | 1e-05 | 8 |

Table 10: Training hyperparameters for **GPT-OSS-20B** on the NumGLUE dataset.

# D  PEFT MODULARITY AND ADAPTER STORAGE

A practical requirement for parameter-efficient fine-tuning is that task-specific updates can be stored, swapped, and reloaded as lightweight adapters rather than full model copies. In GASDU, this is achieved by wrapping each trainable layer with a small "delta" parameter block and enabling an explicit adapter export mechanism. After fine-tuning, each wrapped layer emits a compact index–value sparse diff relative to a frozen backbone snapshot, and these diffs can be saved, reloaded, or combined in exactly the same modular way as standard PEFT adapters.

To quantify the footprint of these exported adapters, Table 13 reports the number of trainable parameters and on-disk size for LLaMA-2-7B on NumGLUE Type 1 under a 0.10% update budget. The GASDU diff occupies only a small fraction of the full model (8.4M parameters and 161 MB versus 6.7B parameters and 13,500 MB), making it suitable for multi-task deployment. Although the diff is larger than a minimal LoRA adapter, it reflects the additional flexibility of dynamic sparse updates while preserving the key PEFT advantage of sharing a single backbone across many tasks.

| Model | Method | Upd.% | # params | Disk size (MB) |
|---|---|---|---|---|
| | Full model | 100.0 | 6.7B | 13,500 |
| LLaMA-2-7B | LoRA | 0.10 | 6.3M | 24 |
| | GASDU ($M$=50) | 0.10 | 8.4M | 161 |

Table 11: Adapter size comparison on NumGLUE Type 1 with LLaMA-2-7B under a 0.10% update budget after 3-epoch fine-tuning. For GASDU ($M$=50), we report the number of nonzero entries in the exported sparse diff (including their indices) and the resulting on-disk size. For LoRA, we report the total number of trainable adapter parameters along with its on-disk size. The full LLaMA-2-7B backbone is shown for reference.

# E  ADDITIONAL EXPERIMENTS

## E.1  EFFECT OF THE SPARSITY BUDGET

In the main text we focused on an extreme update regime with 0.01% of parameters updated per wrapped projection, in order to highlight that GASDU can already match or closely approach full fine-tuning while using a very small fraction of weights. To study how performance scales with the sparsity budget $k$, we conduct an additional sweep over three update levels: 0.01%, 0.10%, and 0.50%.

We evaluate LLaMA-3-8B with GASDU ($M$=50) on NumGLUE Types 1–3, using the same training setup as in the main paper (optimizer, number of epochs, and data splits). For each configuration we report the median Exact Match (EM) over three seeds per type, and the average across Types 1–3.

| Model | Method | Update .% | Type1 | Type2 | Type3 | Avg |
|---|---|---|---|---|---|---|
| | | 0.01 | 63.5 | 52.2 | 69.0 | 61.6 |
| LLaMA-3-8B | GASDU ($M = 50$) | 0.10 | 64.2 | 52.5 | 69.5 | 62.1 |
| | | 0.50 | 65.4 | 53.5 | 68.7 | 62.5 |

Table 12: Validation Exact Match (EM, %) on NumGLUE Types 1–3 with LLaMA-3-8B using GASDU ($M$=50) under different update budgets. **Avg** denotes the mean EM across Types 1–3.

The results show that increasing $k$ yields small but consistent gains in EM, which aligns with our theoretical view: a larger budget increases the gradient-retention factor $\alpha_t$, strengthening the PL contraction and improving the robustness of mask reuse as gradients stabilize.

## E.2  GASDU AS PLUG-IN SPARSE ADAPTERS

Although GASDU updates entries in the original weight matrices during training, it does not require storing a separate full model per downstream task. Instead, we implement an explicit *adapter ex-*

*port* mechanism that turns each fine-tuned run into a lightweight, swappable sparse adapter without slowing down the training speed:

- The backbone weights are kept as a frozen reference snapshot.
- Each GASDU-wrapped layer maintains a sparse "delta" parameterization over a budget of $k$ coordinates per update step.
- After training, each layer exports an index–value sparse difference (nonzero coordinates and their values) relative to the frozen backbone.
- These diffs can be saved, reloaded, and composed in the same way as standard PEFT adapters, allowing users to switch tasks without duplicating the full model.

We profile adapter footprints for LLaMA-2-7B under a $0.10\%$ *per-step* update budget on three tasks of increasing size: NumGLUE Type 1 ($\sim 400$ samples), OBQA ($\sim 5K$ samples), and WinoGrande ($\sim 58K$ samples). All adapters (LoRA and GASDU) are exported in `bf16` for a fair comparison.

| Model | Method | Upd.% | # params | Disk size (MB) |
|---|---|---|---|---|
| | Full model | 100.0 | 6.7B | 13,500 |
| | LoRA (all tasks) | 0.10 | 6.3M | 12.1 |
| LLaMA-2-7B | GASDU (NumGLUE Type 1, $\sim 400$ samples) | 0.10 | 8.4M | 48.0 |
| | GASDU (OBQA, $\sim 5K$ samples) | 0.10 | 26.2M | 150.1 |
| | GASDU (WinoGrande, $\sim 58K$ samples) | 0.10 | 52.7M | 301.6 |

Table 13: Adapter footprint for LLaMA-2-7B under a 0.10% per-step update budget. For LoRA, "# params" is the number of trainable adapter parameters in the dense low-rank matrices; the resulting `bf16` checkpoint is essentially task-independent (12.1 MB). For GASDU, "# params" is the number of nonzeros in the exported sparse diff (the union of coordinates updated over training), and the on-disk size grows with dataset size because we store both `bf16` values and integer indices.

Although the GASDU adapters are larger than the minimal LoRA adapters, even on the largest benchmark the $\approx 302\,\text{MB}$ checkpoint is less than $3\%$ of the 13.5 GB backbone, so maintaining many task-specific adapters is still far cheaper than duplicating the base model. Because these sparse diffs reside on CPU/disk and can be loaded or swapped in small chunks, adapter size never becomes a GPU-memory bottleneck, and GASDU preserves the practical plug-in modularity expected of PEFT-style adapters.

### E.3 HUMANEVAL CODE GENERATION RESULTS

To directly address concerns about longer-context and multi-step generation, we additionally evaluate GASDU on the HumanEval code generation benchmark. In this experiment:

- We fine-tune LLaMA-3-8B on the Code-Feedback dataset.
- We evaluate on HumanEval and report PASS@1, computed by executing generated code against the official test cases.
- Due to time and computational constraints, we include full fine-tuning and standard LoRA as baselines.
- For fairness, all methods share a single hyperparameter setting: learning rate $1 \times 10^{-6}$ and batch size $4$, without grid search.

| Model | Method | Upd.% | PASS@1 (HumanEval) |
|---|---|---|---|
| | Full FT | 100.0 | 26.22 |
| LLaMA-3-8B | LoRA | 0.10 | 24.80 |
| | GASDU | 0.10 | 25.61 |

Table 14: PASS@1 (%) on HumanEval for LLaMA-3-8B fine-tuned on Code-Feedback. All methods use the same learning rate ($1 \times 10^{-6}$) and batch size (4).

Under this controlled setting, GASDU with a $0.10\%$ update budget closely matches full fine-tuning and slightly outperforms LoRA on HumanEval, indicating that the proposed dynamic sparse update

mechanism extends beyond short-form reasoning tasks to longer-context, multi-step code generation.

## F  ADAM OPTIMIZER AND MOMENTUM HANDLING IN GASDU

Our theory is stated for masked gradient descent with a scalar stepsize to isolate the role of the gradient–retention factor, but all reported GASDU experiments use an adaptive optimizer (DeepSpeed's FusedAdam). Importantly, Adam is applied only to a small adapter vector per wrapped layer, not to the full dense backbone, so optimizer memory scales with the sparse budget $k$ instead of the model size.

For each wrapped layer we maintain a fixed adapter vector $\delta \in \mathbb{R}^k$ (implemented as `delta_vals`) and keep the dense weight matrix $W$ frozen during the optimizer step. Let $j \in \{1, \ldots, k\}$ index slots of $\delta$, and let $\pi_t(j)$ denote the dense weight coordinate assigned to slot $j$ at step $t$ by the current Top-$k$ mask. Adam is run in the usual way on the slot gradients $g_t^{(j)}$,

$$m_{t+1}^{(j)} \;=\; \beta_1 m_t^{(j)} + (1-\beta_1)g_t^{(j)}, \qquad v_{t+1}^{(j)} \;=\; \beta_2 v_t^{(j)} + (1-\beta_2)\big(g_t^{(j)}\big)^2,$$

$$\delta_{t+1}^{(j)} \;=\; \delta_t^{(j)} - \eta \frac{m_{t+1}^{(j)}}{\sqrt{v_{t+1}^{(j)}} + \varepsilon},$$

where $(m_t^{(j)}, v_t^{(j)})$ are Adam's first and second moments for slot $j$. After each optimizer step we *commit* the adapter updates to the backbone and reset the slots:

$$W_{t+1}\big[\pi_t(j)\big] \;=\; W_t\big[\pi_t(j)\big] + \delta_{t+1}^{(j)}, \qquad \delta_{t+1}^{(j)} \leftarrow 0 \quad \forall j,$$

while keeping $(m_{t+1}^{(j)}, v_{t+1}^{(j)})$ intact.

When the mask is refreshed every $M$ steps (including $M=1$), the assignment $\pi_t$ changes and a slot $j$ may be reassigned from a coordinate $u$ at step $t$ to a new coordinate $v$ at step $t+1$. In that case, the first update to $v$ uses a "warm-start" state $(m_{t+1}^{(j)}, v_{t+1}^{(j)})$ inherited from $u$. Empirically, this slot-based momentum trains stably and $M=1$ consistently matches or outperforms larger $M$, indicating that such warm starts do not create optimization pathologies in practice.

Conceptually, Adam is attached to the $k$ adapter slots rather than to fixed dense coordinates, so it is best viewed as providing an adaptive step-size schedule in a $k$-dimensional adapter space, not as exact per-coordinate momentum on all weights. The optimizer state tensors $(m_t, v_t)$ therefore have the same fixed shape as $\delta$ (two length-$k$ vectors per wrapped layer), so the optimizer memory is $\mathcal{O}(k)$ and does not grow with the number of distinct dense coordinates visited by the Top-$k$ selector. This differs fundamentally from "fused SGD," which still computes dense gradients and updates all coordinates, whereas GASDU changes *which* coordinates are ever updated and stores optimizer state only for that sparse subset.

