# OpenReview forum: "GASDU: Gauss--Southwell Dynamic Update for Efficient LLM Fine-Tuning"
_ICLR.cc/2026/Conference — Submitted to ICLR 2026_

### Official Review · Reviewer_tmdL · 2025-10-29

**Soundness:** 4
**Presentation:** 3
**Contribution:** 3
**Rating:** 8
**Confidence:** 4

**Summary:**

The paper proposes a parameter-efficient fine-tuning (PEFT) method for large language models (LLMs) that updates only a dynamically selected subset of parameters. Instead of using static masks or low-rank adapters or incurring high overhead from dynamic sparse methods, GASDU periodically performs Gauss–Southwell–k selection. Linear convergence is proved under a local Polyak–Łojasiewicz (PL)
condition. Empirically, GASDU is shown to match full fine-tuning using only 0.01% of parameters, achieving speedup and memory savings.

**Strengths:**

- The paper is well written. Anonymous reproducibility repository provided.

- The use of Gauss–Southwell-$k$ coordinate selection in a PEFT context is original and well-motivated.

- Provides a clean convergence proof under a local PL condition, which introduces the gradient retention factor, a useful measurable diagnostic linking sparsity, update cadence, and convergence rate.

- Benchmarks span arithmetic reasoning and commonsense tasks. The ablation on refresh period $M$ is convincing.

**Weaknesses:**

- Only 0.01% update budget tested; performance across larger budgets would help understand scaling.

- The authors could clarify the effect of $k$.

- On line 777, one citation is not properly rendered.

**Questions:**

- Theoretically, how does the choice of $k$ impact the convergence behavior of the proposed algorithm?

---

> ### Author Response · Authors · 2025-11-20
> **Response to Reviewer tmdL**
>
> We sincerely thank the reviewer for the detailed and constructive feedback, as well as the positive evaluation of GASDU’s soundness, presentation, and contribution. Below, we provide a point-to-point response to each weakness and question raised in the review.
>
> **Response to Weakness 1: Effect of sparsity budget $k$.**
> We thank the reviewer for this valuable comment. To evaluate GASDU under larger sparsity budgets and clarify the effect of the sparsity budget $k$, we have added experiments using multiple budgets (update percentage in {0.01%, 0.10%, 0.50%}). Results for LLaMA-3-8B on NumGLUE Types 1–3 are shown in the table below. We observe that the average EM improves from $61.6$ to $62.1$ to $62.5$ as $k$ increases, indicating small but consistent gains that align with our PL-based interpretation that larger $k$ yields higher gradient-retention factors and slightly faster convergence. One interesting and valuable future direction is to explore the optimal choice of $k$ that balances the budget constraint with performance requirements.
>
> | Model      | Method         | Upd.% | Type1 | Type2 | Type3 |  Avg |
> | ---------- | -------------- | :---: | :---: | :---: | :---: | :--: |
> | LLaMA-3-8B | GASDU ($M=50$) |  0.01 |  63.5 |  52.2 |  69.0 | 61.6 |
> | LLaMA-3-8B | GASDU ($M=50$) |  0.10 |  64.2 |  52.5 |  69.5 | 62.1 |
> | LLaMA-3-8B | GASDU ($M=50$) |  0.50 |  65.4 |  53.5 |  68.7 | 62.5 |
>
> *Table: Validation Exact Match (EM, %) on NumGLUE Types 1–3 with LLaMA-3-8B using GASDU ($M=50$) under different update budgets. For each configuration, we report the median EM over three seeds for each type; **Avg** denotes the mean EM across Types 1–3.*
>
> **Response to Q1: On the theoretical effect of $k$.**
> We thank the reviewer for this thoughtful question. Theoretically, in our masked-gradient view, GASDU’s convergence rate under the local PL condition is
> $$
> 1 - \alpha_t \mu \gamma,
> $$
> where the gradient-retention factor
> $$
> \alpha_t = \frac{\|\Lambda^{(t)} \odot \nabla f(W^{(t)})\|_2^2}
>                {\|\nabla f(W^{(t)})\|_2^2}
> $$
>
>
> is non-decreasing in the Top-$k$ budget, so increasing $k$ drives $\alpha_t$ toward $1$ and makes the rate approach that of full gradient descent (Theorem 4.2). Our mask-reuse analysis (Theorem 4.3) further shows that the lower bound on $\alpha_t$ over a refresh window depends monotonically on the initia $\alpha_{t_{\mathrm{ref}}}$ and decays only sublinearly with the reuse length, so larger $k$ improves both the per-step contraction and robustness to stale masks for a fixed step size and refresh period. We have added these clarifications, with the new explanatory sentences highlighted in **red** in the theoretical analysis section, to more explicitly connect the choice of $k$ to the convergence behavior of GASDU.

---

### Official Review · Reviewer_RGWn · 2025-10-30

**Soundness:** 3
**Presentation:** 3
**Contribution:** 2
**Rating:** 4
**Confidence:** 5

**Summary:**

This paper proposes a PEFT method based on dynamic sparse updates, i.e., dynamically selecting the top-k largest-magnitude gradients for updates and retaining/refreshing the selected mask every M steps. The paper provides both theoretical analysis and an efficient implementation of the method. The approach is evaluated on LLaMA and GPT-OSS models, showing improved fine-tuning accuracy and comparable throughput.

**Strengths:**

1). The paper is well written, and the proposed method is reasonably novel.

2). It provides theoretical proofs that the proposed sparse update achieves a linear convergence rate and that the masks can be reused during each refresh period.

3). The paper presents an efficient implementation that avoids materializing the full gradient matrix, achieving throughput comparable to other PEFT methods.

**Weaknesses:**

1). Unlike LoRA, the proposed dynamic update method appears to require one full model per downstream task, which limits its scalability for multi-task deployment in practice.

2). The baselines used for comparison are relatively weak. The paper relies on basic baseline settings, whereas both LoRA-based and fixed-mask methods have advanced significantly over the past year (see a few selected references below). Without comparison to state-of-the-art methods, it is difficult to assess the true advantages of the proposed approach.

References:

•	LoRA-One: http://arxiv.org/abs/2502.01235

•	LoRA-Pro: http://arxiv.org/abs/2407.18242

•	LoRA-GA: http://arxiv.org/abs/2407.05000

•	SMT: https://openreview.net/forum?id=GbgCRJedQ7

•	Diablo: http://arxiv.org/abs/2506.03230

**Questions:**

In addition to the weaknesses noted above:

1). In Table 3, the ablation on M shows large variance—results for M between 1 and 100 appear to have little impact and show no clear trend. Can the authors provide an explanation?

2). Is the proposed method (GASDU) implemented with DeepSpeed’s FusedAdam optimizer? If so, how are the first and second moments handled?

3). Can this algorithm be applied in tensor-parallel or fully sharded data-parallel settings?

---

> ### Author Response · Authors · 2025-11-20
>
> **Response to Reviewer RGWn**
>
> We thank the reviewer for the careful reading, constructive feedback, and helpful references. Below we respond to the main weaknesses and questions point by point, and we will incorporate these clarifications and additional experiments into the revised version.
>
> ---
>
> **Response to Weakness 1: Clarification on PEFT modularity and adapter storage.**
> Thank you for raising this point, and we apologize for the earlier confusion. Although GASDU updates entries in the original weight matrices during training, it does **not** require storing a separate full model per task. Instead, as described in the new appendix section “GASDU as Plug-In Sparse Adapters,” we keep the backbone as a frozen reference and equip each GASDU-wrapped layer with a sparse “delta’’ over a budget of $k$ coordinates per step. After training, each layer exports an index–value sparse diff relative to the backbone, which can be saved, reloaded, and composed in the same way as standard PEFT adapters.
>
> To make the footprint concrete, we profile adapter sizes for LLaMA-2-7B under a fixed (0.10%) per-step update budget across three tasks, exporting all adapters in `bf16`:
>
> | **Model**  | **Method**                                 | **Upd.%** | **# params** | **Disk size (MB)** |
> | ---------- | ------------------------------------------ | --------- | ------------ | ------------------ |
> | LLaMA-2-7B | Full model                                 | 100.0     | 6.7B         | 13,500             |
> | LLaMA-2-7B | LoRA (all tasks)                           | 0.10      | 6.3M         | 12.1               |
> | LLaMA-2-7B | GASDU (NumGLUE Type 1, $\sim 400$ samples) | 0.10      | 8.4M         | 48.0               |
> | LLaMA-2-7B | GASDU (OBQA, $\sim 5$K samples)            | 0.10      | 26.2M        | 150.1              |
> | LLaMA-2-7B | GASDU (WinoGrande, $\sim 58$K samples)     | 0.10      | 52.7M        | 301.6              |
>
> *Table 1: Adapter footprint for LLaMA-2-7B under a 0.10% per-step update budget. For LoRA, `#params` is the number of trainable adapter parameters in the dense low-rank adapters. For GASDU, `#params` is the number of nonzeros in the exported sparse diff (union over training steps). All adapters are stored in `bf16`.*
>
> While GASDU adapters are larger than minimal LoRA adapters, even the largest ($\approx 302$ MB) is <3% of the 13.5 GB backbone, so maintaining many task-specific adapters is still far cheaper than duplicating the base model. Since these sparse diffs live on CPU/disk and are swapped in as needed, they do not create a GPU-memory bottleneck and preserve the plug-in modularity expected of PEFT-style adapters.
>
>
> ---
>
> **Response to Weakness 2: Comparison with recent LoRA variants.**
> Thank you for the valuable suggestion. We have now added LoRA-One and LoRA-GA as strong recent LoRA-family baselines on the NumGLUE arithmetic benchmarks under the same 0.01% update budget as our other PEFT methods. On NumGLUE, GASDU consistently matches or outperforms these variants across all three backbones (LLaMA-2-7B, LLaMA-3-8B, GPT-OSS-20B). The corresponding LoRA-One/LoRA-GA experiments on the commonsense benchmark suite are currently in production, and we will update the main tables once these results are finalized.
>
> | Model       | Method   | Upd. % | type1    | type2    | type3    | type4    | type5    | type8    | Avg      |
> | ----------- | -------- | ------ | -------- | -------- | -------- | -------- | -------- | -------- | -------- |
> | LLaMA-2-7B  | LoRA-One | 0.01   | 28.4     | 38.5     | 51.2     | 45.0     | 30.4     | 31.5     | 37.5     |
> | LLaMA-2-7B  | LoRA-GA  | 0.01   | 29.6     | 40.3     | 51.2     | **49.1** | 31.3     | 29.1     | 38.4     |
> | LLaMA-2-7B  | GASDU    | 0.01   | **31.4** | **42.7** | *51.7*   | *47.8*   | **33.5** | **34.4** | **40.2** |
> | LLaMA-3-8B  | LoRA-One | 0.01   | 60.5     | 48.6     | 62.3     | 57.7     | 66.1     | 46.5     | 57.0     |
> | LLaMA-3-8B  | LoRA-GA  | 0.01   | 61.7     | 50.5     | *70.4*   | *60.9*   | 61.4     | *48.4*   | 58.9     |
> | LLaMA-3-8B  | GASDU    | 0.01   | **63.5** | *52.2*   | 69.0     | **62.3** | *67.9*   | **50.1** | *60.8*   |
> | GPT-OSS-20B | LoRA-One | 0.01   | 70.4     | **52.0** | 63.0     | **68.2** | **71.1** | **53.1** | 63.0     |
> | GPT-OSS-20B | LoRA-GA  | 0.01   | **74.3** | 50.7     | *80.5*   | *68.1*   | 67.2     | 50.6     | *65.2*   |
> | GPT-OSS-20B | GASDU    | 0.01   | *73.3*   | *51.1*   | **81.4** | 66.4     | 69.9     | *52.9*   | **65.8** |
>
> *Table 2: NumGLUE arithmetic reasoning results for advanced LoRA variants and GASDU (best in **bold**, second-best in *italics*). All experiments follow the same setup as in the main paper.*
>
> ---

---

> > ### Author Response · Authors · 2025-11-20
> >
> > ### 3. Response to specific questions
> >
> > **Response to Q1: Ablation on $M$ (mask refresh interval) and variance in Table 3.**
> > We agree that Table 3 shows relatively small differences across $M \in [1,100]$ and no clear monotone trend. Our view is that **mask reuse is effective over a wide range of $M$**. With three epochs of training (thousands of iterations), Figure 4 in the main text shows that the top-$k$ masks stabilize and change little in the later stage (after roughly the first 800 iterations). As a result, for moderate values of $M$ the same informative mask can be reused for many steps without hurting convergence, so different choices of $M$ in this range lead to similar final EM/accuracy.
> >
> > **Response to Q2: Optimizer and Adam moments.**
> > Yes. All experiments related to GASDU are conducted with FusedAdam. Before fine-tuning, we freeze all original dense weights in the backbone, and only the sparse update vectors $\delta_{\text{vals}}$ are marked as trainable. Consequently, Adam’s first and second moments are maintained *only* for these small trainable vectors; no momentum state is allocated for the frozen base weights or for the mask indices. After each optimizer step, the current sparse updates in $\delta_{\text{vals}}$ are folded into the base weights and $\delta_{\text{vals}}$ is reset to zero, while the associated Adam moment buffers remain attached to the same parameter slots and are reused in subsequent steps. Thus, the memory cost of the first and second moments scales with $O(k)$ per wrapped projection (plus biases), rather than with the full parameter count of the model.
> >
> > **Response to Q3: Applicability to tensor-parallel / fully sharded data-parallel.**
> > GASDU is compatible with tensor-parallel (TP) and fully sharded data-parallel (FSDP) architectures because it only augments local linear projections with a small sparse update vector and per-layer mask indices, and all its computations (streaming Top-$k$, sparse updates, commit) are defined on the local activations and gradients. In a TP setup, each shard can independently apply GASDU to its partition of the projection matrix, selecting Top-$k$ entries and updating them within that shard, exactly as is standard for applying LoRA-style adapters per shard. In an FSDP setting, the sparse updates are simply additional trainable parameters that can be sharded or replicated while the frozen base weights are sharded as usual, so extending GASDU to TP/FSDP mainly requires standard integration into the parallelization framework rather than any change to the algorithm itself. In the revised manuscript, we have added a clarification paragraph (“Parallel training compatibility”) in the Method section explicitly explaining that GASDU is compatible with both TP and FSDP.

---

### Official Review · Reviewer_VzyC · 2025-10-31

**Soundness:** 3
**Presentation:** 3
**Contribution:** 2
**Rating:** 2
**Confidence:** 5

**Summary:**

The paper proposes GASDU, a dynamic sparse fine-tuning method that periodically selects top-k gradient coordinates (Gauss–Southwell rule) and updates only those parameters, reusing the mask for several steps. It provides convergence analysis under a local PL condition and conducted numerical experiments.

**Strengths:**

Simple and well-motivated dynamic sparse update rule.
Clear theoretical analysis with convergence guarantees under a PL assumption.

**Weaknesses:**

1. The proposed method appears to work only with plain SGD. When combined with adaptive optimizers such as Adam, the dynamic masking conflicts with momentum updates: maintaining moment estimates for all parameters contradicts the claimed memory efficiency, while periodically resetting them typically leads to instability and poor convergence. If only plain SGD is used, there is little need for a separate algorithm—SGD updates can be directly fused into backpropagation (i.e., updating parameters as gradients are computed and then clearing them), which essentially replicates the proposed masking behavior.
2. The paper lacks comparisons with recent LoRA variants (e.g., LoRA-One, LoRA-GA, MiLoRA), which substantially improve fine-tuning quality and represent stronger baselines.
3. Because the proposed algorithm updates entries across the original weight matrices rather than using lightweight adapters, it loses one of the key benefits of PEFT—the ability to store and switch between multiple small task-specific adapters. GASDU cannot easily support such modularity.
4. The theoretical analysis mainly restates the classical PL convergence result with minor modifications for a masked gradient. The provided theory does not offer clear insight into why the proposed method should outperform existing approaches or how it explains the observed empirical gains.

**Questions:**

See weakness.

---

> ### Author Response · Authors · 2025-11-20
> **Response to Reviewer VzyC**
>
> We thank Reviewer VzyC for the thoughtful and detailed feedback. Below, we provide point-by-point responses to each weakness raised in the review.
>
> **Response to Weakness 1: On optimizer choice, momentum, and relation to fused SGD.**
> We thank the reviewer for raising these questions and believe there is a slight misunderstanding here: while our theory is stated for masked gradient descent with a scalar stepsize to isolate the role of the gradient retention factor, our implementation is *not* restricted to plain SGD. All reported experiments related to GASDU use FusedAdam on a small vector of delta coefficients per wrapped layer with the dense backbone frozen. The adaptive optimizer (and its moment buffers) is applied only to these low-dimensional delta parameters and biases, so optimizer memory scales with the sparse budget $k$ rather than full model size, and we never maintain or reset dense moments for all weights. This differs fundamentally from “fused SGD,” which still computes dense gradients and updates all coordinates, whereas GASDU changes *which* coordinates are ever updated and keeps optimizer state only for that sparse subset. Extending the PL-style analysis to fully adaptive optimizers is an interesting direction for future work.
>
> **Response to Weakness 2: Comparison with recent LoRA variants.**
> Thank you for the valuable suggestion. We have added LoRA-One and LoRA-GA as recent LoRA-family baselines on the NumGLUE arithmetic benchmarks under the same 0.01% update budget as other PEFT methods. On NumGLUE, GASDU consistently matches or outperforms these variants across all three backbones (LLaMA-2-7B, LLaMA-3-8B, GPT-OSS-20B). The corresponding LoRA-One/LoRA-GA experiments on the commonsense benchmark suite are currently in production, and we will update the main tables once these results are finalized.
>
> | Model       | Method   | Upd. % | type1    | type2    | type3    | type4    | type5    | type8    | Avg      |
> | ----------- | -------- | ------ | -------- | -------- | -------- | -------- | -------- | -------- | -------- |
> | LLaMA-2-7B  | LoRA-One | 0.01   | 28.4     | 38.5     | 51.2     | 45.0     | 30.4     | 31.5     | 37.5     |
> | LLaMA-2-7B  | LoRA-GA  | 0.01   | 29.6     | 40.3     | 51.2     | **49.1** | 31.3     | 29.1     | 38.4     |
> | LLaMA-2-7B  | GASDU    | 0.01   | **31.4** | **42.7** | *51.7*   | *47.8*   | **33.5** | **34.4** | **40.2** |
> | LLaMA-3-8B  | LoRA-One | 0.01   | 60.5     | 48.6     | 62.3     | 57.7     | 66.1     | 46.5     | 57.0     |
> | LLaMA-3-8B  | LoRA-GA  | 0.01   | 61.7     | 50.5     | *70.4*   | *60.9*   | 61.4     | *48.4*   | 58.9     |
> | LLaMA-3-8B  | GASDU    | 0.01   | **63.5** | *52.2*   | 69.0     | **62.3** | *67.9*   | **50.1** | *60.8*   |
> | GPT-OSS-20B | LoRA-One | 0.01   | 70.4     | **52.0** | 63.0     | **68.2** | **71.1** | **53.1** | 63.0     |
> | GPT-OSS-20B | LoRA-GA  | 0.01   | **74.3** | 50.7     | *80.5*   | *68.1*   | 67.2     | 50.6     | *65.2*   |
> | GPT-OSS-20B | GASDU    | 0.01   | *73.3*   | *51.1*   | **81.4** | 66.4     | 69.9     | *52.9*   | **65.8** |
>
> *Table 1: NumGLUE arithmetic reasoning results for advanced LoRA variants and GASDU (best in **bold**, second-best in *italics*). All experiments follow the same setup as in the main paper.*
>
> ---

---

> > ### Comment · Reviewer_VzyC · 2025-11-24
> >
> > Thank you very much for the responses and revisions. I appreciate the explanations the additional experiments. However, several important issues remain unclear and should be addressed for the paper.
> >
> > ## 1. Adam Optimizer and Momentum Handling
> >
> > The paper still does not specify how Adam's momentum states are handled under dynamic sparse updates. This point is critical because the behavior of Adam directly affects optimization stability and fairness of comparison.
> >
> > ### (a) If momentum is *not* retained across refresh steps
> >
> > - Reinitializing Adam's first and second moments every time the sparse set changes can significantly destabilize training, as momentum accumulation is central to Adam.
> > - Additionally, when M = 1, the method essentially becomes Sign-SGD without momentum. However, Table 3 shows that M = 1 yields the best performance.
> >   This raises the question: Why wouldn't simple SGD or Sign-SGD over the full parameter set achieve similar or better results?
> >
> > ### (b) If momentum *is* retained for parameters that are no longer active
> >
> > This introduces several issues:
> >
> > - If a parameter is active for steps \( t_1 - s_1 \), inactive for \( t_2 - s_2 \), and active again at \( t_3 - s_3 \), how is the iteration count determined? Are the momentum terms accumulated during the inactive period?
> >
> > - GASDU continually selects new coordinates as training progresses. If Adam states for all previously selected parameters are retained, the optimizer state memory grows throughout training, making the comparison with other PEFTs unfair especially under the tested extremely restricted paramter budget.
> >
> > - Since GASDU does not know ahead of time how many coordinates will be selected over training, the final optimizer memory footprint is unpredictable—problematic for resource-constrained devices. For instance, Table 4(a) reports that GASDU uses more memory than the fixed mask on the ARC-C dataset, which is tiny. Will the difference be larger for finetuning on larger tasks?
> >
> > These details significantly affect reproducibility and the validity of the claimed benefits, and should be explicitly clarified in the paper.
> >
> > ## 2. Comparison With SOTA PEFT Methods
> >
> > Thank you for adding the new results.

---

> > > ### Comment · Reviewer_VzyC · 2025-11-24
> > >
> > > ## 3. Adapter Storage and Table Interpretation
> > >
> > > In Table 3, GASDU requires substantially more disk space than LoRA to store the adapter. Since this experiment is conducted only on the small NumGLUE Type 1 dataset (about 400 samples), I would expect the adapter size for GASDU to grow even more on standard, larger datasets. It would be helpful if the authors could also report adapter sizes on larger tasks, e.g., the commonsense reasoning.
> > >
> > > Is there something wrong with the table? Lora with 6.3M parameters uses 24MB disk space, while GASDU with 8.4MB parameters uses 161MB disk space. Does your #params mean the number of trainable parameters in your sparse set? If that is the case, why is Upd. % for both LoRA and GASDU the same?
> > >
> > > ## 4. Theoretical Insights
> > >
> > > Thank you for the clarification! The theoretical result can serve as an addition once the major issues outlined above are fully resolved.

---

> > > > ### Author Response · Authors · 2025-11-25
> > > >
> > > > **Response to Comment 3.**
> > > >
> > > > We thank the reviewer for carefully examining the table and raising concerns about the reported adapter sizes. First, the original table is numerically correct; the seemingly disproportionate disk footprint for GASDU arises because, in addition to storing the sparse diff values, we also store explicit index tensors (in 32-bit format) for every nonzero entry, whereas LoRA only stores dense adapter values. To ensure a fair comparison, we now export all adapters (LoRA and GASDU) in `bf16` and re-ran the profiling on small-, medium-, and large-scale tasks.
> > > >
> > > > **Meaning of `#params` and `Upd.%`.**
> > > > For LoRA, `#params` is the number of trainable adapter weights in the dense low-rank matrices (about $6.3$M), while for GASDU it is the number of nonzero entries in the final exported sparse diff, i.e., the union of all coordinates that were significantly updated during training. `Upd.%` is a *per-step* budget (0.10% of backbone parameters) used to set either the LoRA rank or the GASDU per-layer $k$, so LoRA and GASDU have the same training-time sparsity even though the cumulative union of GASDU updates can exceed the per-step budget.
> > > >
> > > > **Disk space and task size.**
> > > > LoRA’s disk usage is essentially fixed across tasks because the adapter structure (rank and target modules) is fixed, giving a `bf16` checkpoint of about 12.1 MB. GASDU’s disk usage grows with dataset size because the sparse diff records the union of updated coordinates together with their indices, leading to `bf16` checkpoints of about 48.0 MB on NumGLUE Type 1 ($\sim 400$ samples), 150.1 MB on OBQA ($\sim 5$K samples), and 301.6 MB on WinoGrande ($\sim 58$K samples), all of which remain far smaller than storing a full 13.5 GB backbone per task. Even on the largest benchmark, the $\approx 302$ MB GASDU adapter is less than $3%$ of the backbone size, so maintaining many such task-specific adapters is still much cheaper than duplicating the base model. Because these adapters reside on CPU/disk and can be loaded or swapped in small chunks, adapter size never becomes a GPU-memory bottleneck, and GASDU therefore preserves the practical modularity that motivates PEFT-style methods. We have updated “E.2 GASDU AS PLUG-IN SPARSE ADAPTERS” in the appendix accordingly.
> > > >
> > > > | **Model**  | **Method**                                 | **Upd.%** | **# params** | **Disk size (MB)** |
> > > > | ---------- | ------------------------------------------ | --------- | ------------ | ------------------ |
> > > > | LLaMA-2-7B | Full model                                 | 100.0     | 6.7B         | 13,500             |
> > > > | LLaMA-2-7B | LoRA (all tasks)                           | 0.10      | 6.3M         | 12.1               |
> > > > | LLaMA-2-7B | GASDU (NumGLUE Type 1, $\sim 400$ samples) | 0.10      | 8.4M         | 48.0               |
> > > > | LLaMA-2-7B | GASDU (OBQA, $\sim 5$K samples)            | 0.10      | 26.2M        | 150.1              |
> > > > | LLaMA-2-7B | GASDU (WinoGrande, $\sim 58$K samples)     | 0.10      | 52.7M        | 301.6              |
> > > >
> > > > *Table 1: Adapter footprint for LLaMA-2-7B under a 0.10% per-step update budget. For LoRA, `#params` is the number of trainable adapter parameters in the dense low-rank adapters. For GASDU, `#params` is the number of nonzeros in the exported sparse diff (union over training steps). All adapters are stored in `bf16`.*

---

> > > ### Author Response · Authors · 2025-11-25
> > >
> > > **Response to Comment 1.**
> > > We thank the reviewer for this thoughtful comment and the opportunity to clarify our optimizer design. We first clarify that our implementation matches neither case (a) nor case (b) as posed in the comment. We do *not* reinitialize Adam’s first and second moments when the sparse mask is refreshed, but we also do *not* maintain separate momentum buffers for every dense weight coordinate that has ever been selected. Instead, for each wrapped layer we maintain a fixed adapter vector $\delta \in \mathbb{R}^k$ (implemented as `delta_vals`) and apply Adam only to this vector (plus biases). Let $j \in {1,\dots,k}$ index slots of $\delta$ and let $\pi_t(j)$ denote the dense weight coordinate assigned to slot $j$ at step $t$ by the current Top-$k$ mask. Adam is run in the usual way on the slot gradients $g_t^{(j)}$,
> > > $$
> > > m_{t+1}^{(j)} = \beta_1 m_t^{(j)} + (1-\beta_1) g_t^{(j)},
> > > \qquad
> > > v_{t+1}^{(j)} = \beta_2 v_t^{(j)} + (1-\beta_2) \bigl(g_t^{(j)}\bigr)^2,
> > > $$
> > > $$
> > > \delta_{t+1}^{(j)} = \delta_t^{(j)} - \eta ,\frac{m_{t+1}^{(j)}}{\sqrt{v_{t+1}^{(j)}} + \varepsilon},
> > > $$
> > > and after each optimizer step we *commit* the adapter update to the frozen backbone and reset the slots via
> > > $$
> > > W_{t+1}\bigl[\pi_t(j)\bigr] = W_t\bigl[\pi_t(j)\bigr] + \delta_{t+1}^{(j)},
> > > \qquad
> > > \delta_{t+1}^{(j)} \leftarrow 0 \quad \forall j,
> > > $$
> > > while keeping $(m_{t+1}^{(j)}, v_{t+1}^{(j)})$ intact. When $M=1$, the mask is recomputed every step, so the assignment $\pi_t$ changes and a slot $j$ may be reassigned from a coordinate $u$ at step $t$ to a new coordinate $v$ at step $t+1$; the first update to $v$ then uses a “warm-start” state $(m_{t+1}^{(j)}, v_{t+1}^{(j)})$ inherited from $u$. Empirically, across all reported tasks and three random seeds, this slot-based momentum trains stably and the $M=1$ configuration consistently matches or outperforms larger $M$ (Table 3), indicating that this warm start does not introduce optimization pathologies in practice. Conceptually, because Adam is attached to the fixed set of $k$ slots rather than to fixed dense coordinates, it is best viewed as providing an adaptive step-size schedule in a $k$-dimensional adapter space, rather than exact per-coordinate momentum on the full model. Crucially, the optimizer state tensors $(m_t, v_t)$ have the same fixed shape as $\delta$ (two length-$k$ vectors per wrapped layer), so the optimizer memory is $\mathcal{O}(k)$ and fully determined once $k$ is chosen; it does not grow with the number of distinct dense coordinates visited by the Top-$k$ selector during training.

---

> ### Author Response · Authors · 2025-11-20
>
> **Response to Weakness 3: Clarification on PEFT modularity and adapter storage.**
> Thank you very much for raising this point, and we apologize for the earlier confusion. In practice, GASDU does **not** require storing a full model for each downstream task. As in other dynamic-update PEFT methods, the base model is shared and each task contributes only a lightweight task-specific delta. We have modified the implementation of GASDU to allow an explicit adapter export mechanism: after fine-tuning, each wrapped layer outputs a compact index–value sparse diff relative to a frozen backbone snapshot. These diffs can be saved, swapped, or reloaded in the same modular manner as standard PEFT adapters.
>
> To clarify the practical footprint, we provide an adapter size comparison for LLaMA-2-7B on NumGLUE Type 1 in the table below. The exported diff occupies only a small fraction of the parameters (8.4M parameters and 161 MB vs. a 6.7B-parameter, 13,500 MB full model), making it suitable for multi-task deployment. Although the diff is larger than a minimal LoRA adapter, it reflects the flexibility of dynamic sparse updates, preserving the benefit of PEFT while delivering stronger downstream performance. We have added a section (“PEFT Modularity and Adapter Storage”) in the appendix of the revised manuscript to make this point clear. The new implementation can be found in the anonymous repository provided in the paper.
>
> | Model      | Method         | Upd. % | # params | Disk size (MB) |
> | ---------- | -------------- | ------ | -------- | -------------- |
> | LLaMA-2-7B | Full model     | 100.0  | 6.7B     | 13,500         |
> | LLaMA-2-7B | LoRA           | 0.10   | 6.3M     | 24             |
> | LLaMA-2-7B | GASDU ($M=50$) | 0.10   | 8.4M     | 161            |
>
> *Table 2: Adapter size comparison on NumGLUE Type 1 with LLaMA-2-7B under a 0.10% update budget (3-epoch fine-tuning). For GASDU ($M=50$), we report the number of nonzero entries in the exported sparse diff (including their indices) and the resulting on-disk size. For LoRA, we report the total number of trainable adapter parameters along with its on-disk size. The full LLaMA-2-7B backbone is shown for reference.*
>
> ---
>
> **Response to Weakness 4: Theoretical insights.**
> We respectfully disagree that our theory merely restates the classical PL result. Theorem 4.2 shows that, under a local PL condition, GASDU converges linearly with contraction factor $(1 - \alpha_t \mu \gamma)$, where the *measurable* gradient-retention factor $\alpha_t$ is defined as the fraction of $\ell_2$ gradient norm captured by the current mask; this directly links the observed retention trajectory to how sparsity choices (via $k$) slow down or preserve the full-FT rate. The mask-reuse analysis in Theorem 4.3 further provides a lower bound on $\alpha_t$ in terms of the reuse length $\tau_t \le M$ through a drift parameter $\rho_t = \Theta(\sqrt{\gamma \tau_t})$, showing that retention decays only sublinearly within each refresh window and thereby giving principled guidance for choosing moderate refresh periods $M$ that keep $\alpha_t$ high (and convergence close to dense GD) while avoiding overly frequent refreshes. Beyond these PL-based results, Appendix A.1–A.3 develop additional theory: Theorem A.1 gives a bias–variance decomposition in a linearized regime that explains how the dynamically grown support $S_T$ trades approximation error against variance, Theorem A.2 establishes a uniform-stability bound in which the generalization gap is controlled by the cumulative retention-weighted contraction, and Theorem A.3 bounds the Rademacher complexity of sparse linear hypotheses, together showing that our retention-based analysis provides structural insight into both optimization and generalization rather than being a purely formal restatement of classical PL convergence.

---

### Official Review · Reviewer_RecR · 2025-11-01

**Soundness:** 3
**Presentation:** 3
**Contribution:** 3
**Rating:** 8
**Confidence:** 3

**Summary:**

This work introduces a new fine-tuning method that updates only a tiny fraction of model weights by refreshing a sparse mask every few/M steps using a Gauss–Southwell selection. By selecting only top-k gradients, the method retains training efficiency through lower memory and compute costs. The authors further show improved convergence speeds and that the approach can maintain on par performance (or at least better than previous PeFT) with full fine-tuning despite using a subset of the parameters. Along with the performance improvement comes comparable (to PeFT methods) speedup and memory efficiency over full-finetuning.

**Strengths:**

1. Broad model compatibility: GASDU demonstrates consistent effectiveness across several large language model families, including LLaMA-2, LLaMA-3, and GPT-OSS-20B. This cross-architecture success indicates that the method is not tightly coupled to any specific model design, highlighting its robustness and potential for general adoption in diverse fine-tuning scenarios.
2. Theoretical foundation: The method is supported by a convergence analysis under the Polyak–Lojasiewicz condition, providing formal assurance of stable and predictable optimization behavior. By introducing the gradient-retention factor as a measurable quantity, the authors establish a link between theoretical guarantees and empirical performance, strengthening confidence in the method’s reliability.
3. Practical and efficient design: GASDU’s streaming top-k selection mechanism eliminates the need for dense gradients, substantially lowering per-iteration computational cost and memory usage. This design enables the method to achieve performance that lands somewhere between lightweight approaches like LoRA and more resource-intensive full fine-tuning, maintaining both efficiency and accuracy.

**Weaknesses:**

1. The current experiments cover commonsense reasoning tasks. To better assess the generality of GASDU, the authors should also evaluate it on benchmarks involving longer input sequences and contextual dependencies.

2. The discussion of sparsity-based PEFT methods is missing. This work focuses on updates to a sparse selection of parameters, along with some online update refreshing. There are some recent relevant works in the domain of PeFT, such as S2FT (NeurIPS 2025)[1] and SparseLoRA (ICML 2025)[2], Galore[3]. Including these would strengthen the discussion on sparsity and help illustrate the novelty of refresh-based updates on a sparse set of weights.

References:

[1] Xinyu Yang, Jixuan Leng, Geyang Guo, Jiawei Zhao, Ryumei Nakada, Linjun Zhang, Huaxiu Yao, Beidi Chen, "S2FT: Efficient, Scalable and Generalizable LLM Fine-tuning by Structured Sparsity", NeurIPS 2025

[2] Samir Khaki, Xiuyu Li, Junxian Guo, Ligeng Zhu, Chenfeng Xu, Konstantinos N. Plataniotis, Amir Yazdanbakhsh, Kurt Keutzer, Song Han, Zhijian Liu, "SparseLoRA: Accelerating LLM Fine-Tuning with Contextual Sparsity", ICML 2025

[3] Jiawei Zhao, Zhenyu Zhang, Beidi Chen, Zhangyang Wang, Anima Anandkumar, Yuandong Tian, "GaLore: Memory-Efficient LLM Training by Gradient Low-Rank Projection", Arxiv 2024

**Questions:**

1. Since most tested tasks produce short outputs, such as those in commonsense reasoning, it would strengthen the paper to include evaluations on longer-context and multi-step generation tasks -- like MT-Bench for dialogue or HumanEval for code generation, and arithmetic reasoning benchmarks -- to show how well GASDU scales to extended contexts and more complex output generations.

2. Could the authors provide some insights into why the retention factor decreases for the first half of training, followed by a steep increase, and maintains a high value for the remainder of training? Previous works, such as SparseLoRA[2] and STEP[4] have shown that sparsity is more sensitive in the early stages and hence choose to keep it dense (i.e, have full retention ratios on PeFT at the beginning), before introducing sparsity in later stages. However, interestingly, the trend in Figure 4.0 points to a less aggressive sparsity in the later stage.

[4] Yucheng Lu, Shivani Agrawal, Suvinay Subramanian, Oleg Rybakov, Christopher De Sa, Amir Yazdanbakhsh, "STEP: Learning N:M Structured Sparsity Masks from Scratch with Precondition" ICML 2023

---

> ### Author Response · Authors · 2025-11-20
> **Response to Reviewer RecR**
>
> We sincerely thank the reviewer for the detailed and constructive feedback, as well as the positive evaluation of GASDU’s soundness, presentation, and contribution. Below, we provide a point-to-point response to each weakness and question raised in the review.
>
> **Response to Weakness 1 & Question 1: Evaluation beyond commonsense reasoning and longer-context tasks.**
> Thank you for the valuable suggestion. Although our experiments already span commonsense reasoning (BoolQ, PIQA, SIQA, HellaSwag, WinoGrande, ARC-E/C, OBQA) and arithmetic reasoning (NumGLUE Types 1–5 and 8), we agree that evaluating on longer-context and multi-step generation benchmarks would further strengthen the paper. To address this, we additionally evaluated GASDU on the HumanEval code-generation benchmark using LLaMA-3-8B fine-tuned on Code-Feedback. We report PASS@1 and include full fine-tuning and standard LoRA as baselines, using the same hyperparameters for all methods. As shown in Table 1, GASDU closely matches full fine-tuning and outperforms LoRA on this longer-context, multi-step generation task. Experiment details and results are presented in Appendix E.3 (*HumanEval Code Generation Results*) of the revised manuscript.
>
> | Model      | Method  | Update % | PASS@1 (HumanEval) |
> | ---------- | ------- | -------- | ------------------ |
> | LLaMA-3-8B | Full FT | 100.0    | **26.22**          |
> | LLaMA-3-8B | LoRA    | 0.10     | 24.80              |
> | LLaMA-3-8B | GASDU   | 0.10     | *25.61*            |
>
> *Table 1: PASS@1 (%) on HumanEval for LLaMA-3-8B fine-tuned on Code-Feedback. All methods use the same learning rate $1\times 10^{-6}$ and batch size 4.*
>
> **Response to Weakness 2: Further discussion on sparsity-based PEFT methods.**
> Thank you for the helpful suggestion regarding recent sparsity-based PEFT approaches. We have expanded the **Related Work** section to explicitly include the suggested discussion on S²FT, SparseLoRA, and GaLore [1,2,3], and to clarify how these lines of work compare with our refresh-based selective-update mechanism. In summary, we note that S²FT and SparseLoRA [1,2] introduce structured or contextual sparsity to reduce activated parameters, while GaLore [3] applies low-rank gradient compression. These methods focus on sparsifying updates or activations, whereas GASDU introduces a refresh-based strategy that selectively updates a sparse set of parameters over time. These additions strengthen the sparsity discussion and better situate our contribution within the PEFT landscape.
>
> **Response to Question 2: Insights on the retention-factor trajectory in Figure 4.**
> We appreciate this insightful observation and agree that early training is a more “critical” regime, as emphasized by STEP [4] and SparseLoRA [2]. In our case, the initial decrease in retention for large $M$ reflects rapidly changing gradients that make stale masks misaligned, whereas later in training, stabilized gradient statistics allow long mask reuse while still preserving a high retention factor. Thus, although the curve appears U-shaped, we view it as consistent with the dense-to-sparse intuition in STEP [4] and SparseLoRA [2]. A natural extension of GASDU is to make the refresh period adaptive: use a small $M$ early on to track rapidly changing gradients, then gradually increase $M$ as the masks stabilize, further amortizing the refresh cost. We have added this discussion in the “Effect of Refresh Period” section on page 8 of the main text.
>
> **References**
>
> [1] Xinyu Yang, Jixuan Leng, Geyang Guo, Jiawei Zhao, Ryumei Nakada, Linjun Zhang, Huaxiu Yao, Beidi Chen. “S2FT: Efficient, Scalable and Generalizable LLM Fine-tuning by Structured Sparsity.” NeurIPS, 2025.
>
> [2] Samir Khaki, Xiuyu Li, Junxian Guo, Ligeng Zhu, Chenfeng Xu, Konstantinos N. Plataniotis, Amir Yazdanbakhsh, Kurt Keutzer, Song Han, Zhijian Liu. “SparseLoRA: Accelerating LLM Fine-Tuning with Contextual Sparsity.” ICML, 2025.
>
> [3] Jiawei Zhao, Zhenyu Zhang, Beidi Chen, Zhangyang Wang, Anima Anandkumar, Yuandong Tian. “GaLore: Memory-Efficient LLM Training by Gradient Low-Rank Projection.” arXiv, 2024.
>
> [4] Yucheng Lu, Shivani Agrawal, Suvinay Subramanian, Oleg Rybakov, Christopher De Sa, Amir Yazdanbakhsh. “STEP: Learning N:M Structured Sparsity Masks from Scratch with Precondition.” ICML, 2023.

---

### Meta-Review · Area_Chair_TRGn · 2026-01-06

**Summary:**

The paper introduces GASDU, a parameter-efficient fine-tuning method that trains only a tiny, dynamically selected subset of model weights. Using periodic Gauss-Southwell-k selection, the method refreshes the active parameter set based on current gradient magnitudes and reuses it for multiple steps, balancing adaptivity and efficiency. A streaming implementation avoids forming dense gradients, keeping the overhead of mask updates negligible. The authors provide convergence guarantees under a local Polyak-Lojasiewicz condition and show that gradient retention degrades only sublinearly between refreshes, explaining GASDU’s near-full-fine-tuning behavior. Experiments demonstrate that GASDU consistently outperforms strong PEFT baselines and closely matches or exceeds full fine-tuning while achieving substantial memory and compute savings.

**Reviewer Concerns:**

Below, I briefly describe the reviewers' concerns and the authors' replies.

**Reviewer RecR.**

1. The reviewer requested additional experiments involving longer input sequences and stronger contextual dependencies. The authors provided the requested experiments, and I believe this concern was at least partially resolved.

2. The reviewer asked for a more detailed discussion of sparsity-based PEFT methods. The authors added the necessary discussion, which appears to adequately address the concern.

**Reviewer VzyC.**

1. The reviewer expressed concern that the paper focuses on standard SGD without momentum or "adaptive" step-size methods such as Adam. After the initial rebuttal, the reviewer remained uneasy about how momentum is handled. Based on the paper and the authors’ responses, this does not appear to be an issue: Adam’s moment estimates do not need to be reinitialized after a mask refresh. Therefore, I believe this concern was properly addressed.

2. The reviewer noted the lack of comparison with LoRA-One, LoRA-GA, and MiLoRA. The authors added comparisons with LoRA-One and LoRA-GA, and the reviewer explicitly thanked them for these additional results. I therefore assume that this concern is almost fully resolved.

3. The reviewer pointed out that GASDU does not naturally support modularity, i.e., the ability to store and switch between multiple lightweight task-specific adapters. After the first round of responses, the reviewer remained concerned. In my view, the authors addressed this issue only partially: while it is possible to store lightweight task-specific deltas, their size depends strongly on the chosen mask and on how frequently the mask is refreshed. As shown in the tables included in the authors’ responses, GASDU requires substantially more disk space than LoRA. This appears to be a genuine limitation of GASDU, and I believe these tables should be included in the main paper.

4. The reviewer argued that the theoretical analysis largely restates classical results under the PL condition. The authors disagreed and persuaded the reviewer that this was not the case. However, from my perspective, Theorem 4.2 is indeed fairly standard; for example, it follows directly as a corollary of Theorem 6 in [Ajalloeian, Ahmad, and Sebastian U. Stich. On the convergence of SGD with biased gradients. arXiv preprint arXiv:2008.00051 (2020)]. Theorem 4.3 appears to be novel, but the resulting convergence rate is worse than $1 - \frac{\mu^2}{kL^2}$. In contrast, for standard Top-K-GD (admittedly not a practical method), existing results yield a rate of $1 - \frac{\mu}{k L}$, which is significantly better than $1 - \frac{\mu^2}{kL^2}$. This discrepancy is not adequately discussed by the authors.

**Reviewer RGWn.**

1. The reviewer raised a concern very similar to point W3 discussed by Reviewer VzyC. As mentioned above, the authors addressed this concern satisfactorily.

2. The reviewer requested additional comparisons with LoRA-One, LoRA-Pro, LoRA-GA, SMT, and Diablo. In my opinion, this concern was only partially addressed, as the authors provided additional comparisons with LoRA-One and LoRA-GA, but not with the other methods.

**Reviewer tmdL.**

The reviewer noted that the effect of the sparsity budget $k$ was not explored extensively. The authors provided additional numerical results in response, and I believe this concern was fully resolved.

**Reviewer Scores:**

**Reviewer RecR.** The reviewer’s concerns were addressed, although the original score was already 8. Given the remaining issues raised by other reviewers, I do not expect this reviewer to increase their score.

**Reviewer VzyC.** As discussed above, concerns W1 and W2 were adequately addressed, whereas W3 and W4 require further clarification. I therefore expect the reviewer to maintain their original score.

**Reviewer RGWn.** One of the reviewer’s concerns was resolved, while the other was only partially addressed. I believe the reviewer would likewise keep their original score.

**Reviewer tmdL.** The reviewer did not raise serious concerns, and the authors addressed the points appropriately.

**My assessment.** In my view, all reviewers appear to have overlooked the fact that Theorem 4.2 follows directly from existing results, as explained above. Since the reviewers highlighted the theoretical contributions as a strength of the paper, recognizing this issue would likely lead Reviewers RecR and tmdL to lower their scores. In addition, several important concerns raised by Reviewers VzyC and RGWn remain insufficiently addressed. For these reasons, my final recommendation is reject.

---

### Decision · Program_Chairs · 2026-01-26

Reject